# A conserved morphogenetic mechanism for epidermal ensheathment of nociceptive sensory neurites

Nan Jiang[1], Jeffrey P Rasmussen[2†], Joshua A Clanton[2], Marci F Rosenberg[2], Kory P Luedke[1], Mark R Cronan[3], Edward D Parker[4], Hyeon-Jin Kim[5,6], Joshua C Vaughan[5,6], Alvaro Sagasti[2], Jay Z Parrish[1*]

[1]Department of Biology, University of Washington, Seattle, United States; [2]Department of Molecular Cell and Developmental Biology, University of California, Los Angeles, Los Angeles, United States; [3]Department of Molecular Genetics and Microbiology, Duke University, Durham, United States; [4]Department of Opthalmology, University of Washington, Seattle, United States; [5]Department of Chemistry, University of Washington, Seattle, United States; [6]Department of Physiology and Biophysics, University of Washington, Seattle, United States

**Abstract** Interactions between epithelial cells and neurons influence a range of sensory modalities including taste, touch, and smell. Vertebrate and invertebrate epidermal cells ensheath peripheral arbors of somatosensory neurons, including nociceptors, yet the developmental origins and functional roles of this ensheathment are largely unknown. Here, we describe an evolutionarily conserved morphogenetic mechanism for epidermal ensheathment of somatosensory neurites. We found that somatosensory neurons in *Drosophila* and zebrafish induce formation of epidermal sheaths, which wrap neurites of different types of neurons to different extents. Neurites induce formation of plasma membrane phosphatidylinositol 4,5-bisphosphate microdomains at nascent sheaths, followed by a filamentous actin network, and recruitment of junctional proteins that likely form autotypic junctions to seal sheaths. Finally, blocking epidermal sheath formation destabilized dendrite branches and reduced nociceptive sensitivity in *Drosophila*. Epidermal somatosensory neurite ensheathment is thus a deeply conserved cellular process that contributes to the morphogenesis and function of nociceptive sensory neurons.
DOI: https://doi.org/10.7554/eLife.42455.001

*For correspondence:
jzp2@uw.edu

Present address: †Department of Biology, University of Washington, Seattle, United States

Competing interests: The authors declare that no competing interests exist.

## Introduction

The innervation patterns of cutaneous receptors determine our responses to external stimuli. Many types of cutaneous receptors form specialized terminal structures with epithelial cells that contribute to somatosensation (*Owens and Lumpkin, 2014*; *Zimmerman et al., 2014*). For example, some low threshold mechanoreceptor afferents form synapse-like contacts with Merkel cells (*Mihara et al., 1979*), which directly respond to mechanical stress and tune gentle touch responses (*Maksimovic et al., 2014*; *Woo et al., 2014*). Similarly, afferent interactions with radially packed Schwann cell-derived lamellar cells in Pacinian corpuscles facilitate high frequency sensitivity (*Loewenstein and Skalak, 1966*). By contrast, although various types of free nerve endings, including nociceptive C-fibers, course over and insert into keratinocytes, much less is known about the anatomy of keratinocyte-sensory neuron coupling, or the mechanisms by which keratinocytes modulate sensory neuron structure and function. Recent findings that keratinocytes express sensory channels (*Peier et al., 2002*; *Bidaux et al., 2015*; *Chen et al., 2016*), respond to sensory stimuli (*Koizumi et al., 2004*; *Xu et al., 2006*; *Moehring et al., 2018*), release compounds that modulate

**eLife digest** Humans and other animals perceive and interact with the outside world through their sensory nervous system. Nerve cells, acting as the body's 'telegraph wires', convey signals from sensory organs – like the eyes – to the brain, which then processes this information and tells the body how to respond. There are different kinds of sensory nerve cells that carry different types of information, but they all associate closely with the tissues and organs they connect to the brain.

Human skin contains sensory nerve cells, which underpin our senses of touch and pain. There is a highly specialized, complex connection between some of these nerve cells and cells in the skin: the skin cells wrap tightly around the nerve cells' free ends, forming sheath-like structures. This 'ensheathment' process happens in a wide range of animals, including those with a backbone, like fish and humans, and those without, like insects.

Ensheathment is thought to be important for the skin's nerve cells to work properly. Yet it remains unclear how or when these connections first appear. Jiang et al. therefore wanted to determine the developmental origins of ensheathment and to find out if these were also similar in animals with and without backbones.

Experiments using fruit fly and zebrafish embryos revealed that nerve cells, not skin cells, were responsible for forming and maintaining the sheaths. In embryos where groups of sensory nerve cells were selectively killed – either using a laser or by making the cells produce a toxin – ensheathment did not occur. Further studies, using a variety of microscopy techniques, revealed that the molecular machinery required to stabilize the sheaths was similar in both fish and flies, and therefore likely to be conserved across different groups of animals. Removing sheaths in fly embryos led to nerve cells becoming unstable; the animals were also less sensitive to touch. This confirmed that ensheathment was indeed necessary for sensory nerve cells to work properly.

By revealing how ensheathment first emerges, these findings shed new light on how the sensory nervous system develops and how its activity is controlled. In humans, skin cells ensheath the nerve cells responsible for sensing pain. A better understanding of how ensheathments first arise could therefore lead to new avenues for treating chronic pain and related conditions.

DOI: https://doi.org/10.7554/eLife.42455.002

sensory neuron function (*Woolf et al., 1997*; *Koizumi et al., 2004*; *Moehring et al., 2018*), and can drive sensory neuron firing (*Baumbauer et al., 2015*; *Pang et al., 2015*), underscore the importance of understanding the coupling of keratinocytes to sensory neurons.

Anatomical studies have demonstrated that peripheral arbors of some mammalian somatosensory neurons insert into keratinocytes, not just intercalate between them (*Munger, 1965*; *Cauna, 1973*). Several factors have hindered characterization of sensory neuron-keratinocyte interactions in mammalian systems, including region-specific differences in sensory neuron-epidermis interactions (*Kawakami et al., 2001*; *Liu et al., 2014*), a still-growing inventory of neuronal cell types that innervate the epidermis (*Usoskin et al., 2015*; *Nguyen et al., 2017*), and a shortage of markers that label discrete populations of sensory neurons. Peripheral arbors of somatosensory neurons are likewise inserted into epidermal cells in invertebrate and non-mammalian vertebrate model systems, making these promising settings for characterizing epithelial cell-neurite interactions. Notably, portions of *Drosophila melanogaster* larval nociceptive class IV dendrite arborization (da) neuron dendrites and *Danio rerio* (zebrafish) larval trigeminal and Rohon-Beard (RB) sensory axons become ensheathed by epidermal cells (*Han et al., 2012*; *Kim et al., 2012*; *O'Brien et al., 2012*), and studies in these systems have provided insight into the structure and possible function of this epidermal ensheathment of free nerve endings.

*Drosophila* and zebrafish epidermal cells wrap sensory neurites by extending membranes around the entire circumference of the sensory neurite. The wrapping epidermal membranes are tightly apposed to one another and the ensheathed neurites, embedding them inside a mesaxon-like structure (*Whitear and Moate, 1998*; *Han et al., 2012*; *Kim et al., 2012*; *O'Brien et al., 2012*). A similar structure has been documented for ensheathed somatosensory neurites in *Caenorhabditis elegans* and humans (*Cauna, 1973*; *Chalfie and Sulston, 1981*), suggesting that ensheathment by epidermal cells is a conserved feature of sensory endings. The most extensive ultrastructural analysis of these

structures suggests that the sensory neurites can be continuously ensheathed over extended lengths of the arbor, stretching several micrometers or more (*O'Brien et al., 2012*). Structurally, the interaction between keratinocytes and somatosensory neurites is reminiscent of ensheathment of peripheral axons by nonmyelinating Schwann cells in Remak bundles, suggesting that keratinocyte ensheathment may likewise regulate sensory neuron structure (*Chen et al., 2003*) and function (*Orita et al., 2013*; *Faroni et al., 2014*).

Although the extent and distribution of sensory neurite-epidermal ensheathment have not been systematically analyzed, many of the documented instances involve highly branched mechanosensory and/or nociceptive neurons. In *Drosophila*, epidermal ensheathment has been linked to control of branching morphogenesis in two ways. First, nociceptive class IV dendrite arborization (c4da) neurons are largely restricted to a two-dimensional plane along the basal surface of epidermal cells to potentiate contact-dependent repulsion and hence tiling (*Han et al., 2012*; *Kim et al., 2012*). However, portions of c4da neurons are apically shifted and ensheathed inside the epidermis, allowing for dendrites of other da neurons to innervate the unoccupied basal space and hence 'share' the territory (*Tenenbaum et al., 2017*). Second, epidermal ensheathment appears to regulate dendrite branching activity, as mutation of the microRNA *bantam*, which regulates dendrite-epidermis interactions (*Jiang et al., 2014*), or knockdown of *coracle (cora)*, which encodes a band 4.1-related protein required for sheath formation (*Tenenbaum et al., 2017*), each increase dendrite branching. Although these studies provide the first signs that epidermal ensheathement plays key roles in somatosensory neuron development, the cellular basis and functional consequences of this sensory neuron-epidermis coupling remain to be determined.

Here, we characterized the cellular events involved in formation of epidermal ensheathment of somatosensory neurites in *Drosophila* and zebrafish. First, we identified a series of reporters that accumulate at epidermal sites of somatosensory dendrite ensheathment in *Drosophila*, demonstrating that sheaths form at specialized membrane domains and providing markers for in vivo tracking of the sheaths. Remarkably, epidermal sheaths are labeled by similar markers in zebrafish, suggestive of a conserved molecular machinery for ensheathment. Using these reporters, we found that epidermal sheaths in *Drosophila* and zebrafish wrap different types of neurons to different extents and that somatosensory neurons are required for formation and maintenance of epidermal sheaths. Finally, we found that blocking epidermal sheath formation led to exuberant dendrite branching and branch turnover, as well as reduced nociceptive sensitivity in *Drosophila*. Altogether, these studies demonstrate that ensheathment of somatosensory neurons by epidermal cells is a deeply conserved cellular process that plays key roles in the morphogenesis and function of nociceptive sensory neurons.

## Results

### PIP2 in epidermal cells is enriched at sites of *Drosophila* dendrite ensheathment

Recent studies have demonstrated that large portions of *Drosophila* c4da dendrite arbors are ensheathed by the epidermis (*Tenenbaum et al., 2017*; *Jiang et al., 2018*). To gain a high resolution view of ensheathment over extended length scales, we subjected *Drosophila* third instar larvae to serial block-face scanning electron microscopy (SBF-SEM) (*Denk and Horstmann, 2004*). Consistent with prior TEM studies that provided a snapshot of these sheath structures (*Han et al., 2012*; *Kim et al., 2012*; *Jiang et al., 2014*), in individual sections we observed dendrites embedded inside epithelial cells and connected to the basal epithelial surface by thin, tubular invaginations formed by close apposition of epidermal membranes (*Figure 1A*). To determine whether c4da dendrites were continuously ensheathed in these mesaxon-like structures, we followed individual dendrites from the site of insertion into the epidermis through EM volumes of abdominal segments cut into 60-nm sections along the apical-basal axis. We found that dendrites were embedded in epithelial cells over extended distances (often several microns or more), that dendrites were continuously embedded in these mesaxon-like structures with elongated tubular invaginations, and that the epidermal membranes comprising the walls of these tubular invaginations were tightly juxtaposed and electron-dense along their entire length (*Figure 1B and C*). Each of these structural elements was previously described for the ensheathment of peripheral axons by keratinocytes in zebrafish (*O'Brien et al.,*

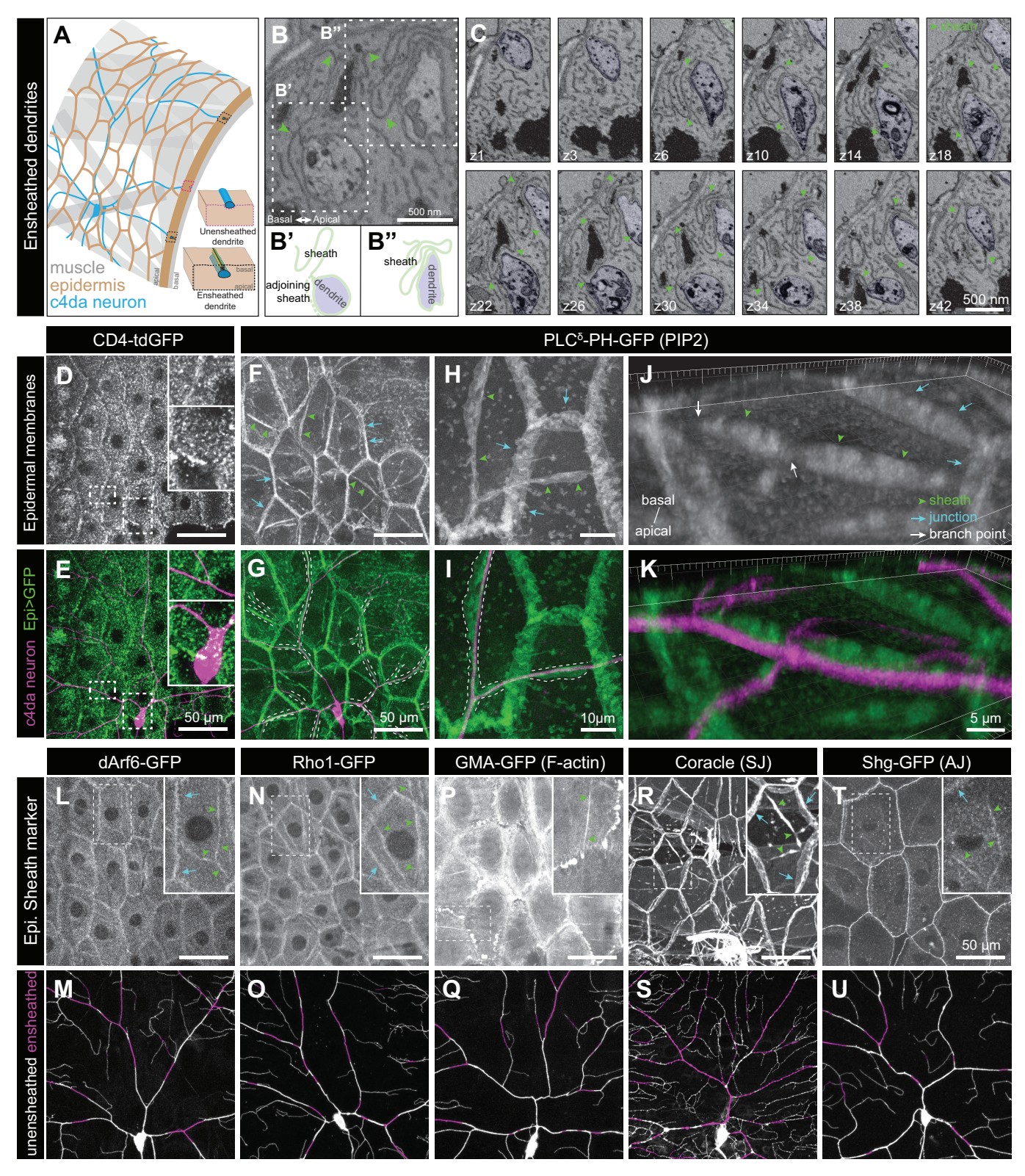

**Figure 1.** Epidermal PIP2 accumulation marks sites of dendrite ensheathment. (**A**) Schematic depicting epidermal neurite ensheathment in the *Drosophila* larval body wall. (**B, C**) SBF-SEM analysis of epidermal dendrite ensheathment. (**B' and B"**) Traces of da neuron dendrites and epidermal sheaths in cross-section. (**C**) Serial sections showing epidermal ensheathment (arrowheads mark sheaths) of da neuron dendrites (shaded green). The dendrite present in sections z1-z38 branches inside an epidermal sheath. See also *Figure 1—video 1*. (**D, E**) Assay for markers of dendrite

*Figure 1 continued on next page*

*Figure 1 continued*

ensheathment. GFP-tagged markers were specifically expressed in the epidermis (*A58-Gal4, Cha-Gal80*) in larvae expressing the c4da-specific marker ppk-CD4-tdTomato. Maximum intensity projections of membrane-targeted CD4-tdGFP (D) and c4da dendrites (E) are shown. Insets show magnified views of c4da dendrites (top) and c4da soma (bottom). (F–K) Epidermal PLC$^\delta$-PH-GFP labels sites of dendrite ensheathment. Maximum intensity projections of epidermal PLC$^\delta$-PH-GFP (F, H, J) and overlay showing PLC$^\delta$-PH-GFP signal in green and ppk-CD4-tdTomato in magenta to label c4da dendrites (G, I, K). Hatched lines mark sheaths. (F–I) XY projections of live confocal images. (J, K) Representative image showing epithelial PIP2 distribution at sites of c4da dendrite contact visualized using expansion microscopy. Image shows a side view of a single epithelial cell and ensheathed c4da dendrites oriented along the apical-basal axis (apical, top). Note the discontinuities in the epithelial sheath at the dendrite branch point and at epithelial intracellular junctions (arrowheads). Sheaths from six independent neurons analyzed with expansion microscopy showed similar structures. Scale bars have been divided by the measured expansion factor of ~4 × and therefore refers to pre-expansion dimensions. (L–U) Epidermal sheath markers. Maximum intensity projections show the distribution of the indicated GFP reporters in the epidermis of 120 h after egg laying (AEL) larvae and composites show portions of c4da dendrite arbors (shaded purple) wrapped by sheaths labeled by the GFP reporters. Experimental genotypes are detailed in *Supplementary file 2*.

DOI: https://doi.org/10.7554/eLife.42455.003

The following video and figure supplements are available for figure 1:

**Figure supplement 1.** Screen for epithelial markers that accumulate at sites of c4da dendrite contact.

DOI: https://doi.org/10.7554/eLife.42455.004

**Figure supplement 2.** c4da neurons are enclosed by epidermal sheaths.

DOI: https://doi.org/10.7554/eLife.42455.005

**Figure 1—video 1.** SBF-SEM analysis of epidermal dendrite ensheathment.

DOI: https://doi.org/10.7554/eLife.42455.006

*2012*), suggesting that the mechanism of epidermal somatosensory neuron ensheathment may be conserved between invertebrates and vertebrates.

We hypothesized that formation of dendrite sheaths likely involves recruitment of factors that create specialized membrane domains. To identify epithelial membrane-associated markers that preferentially localize to sites of dendrite ensheathment, we used the Gal4-UAS system to selectively express GFP-tagged markers in the epidermis of *Drosophila* larvae also expressing the c4da-specific marker *ppk-CD4-tdTomato* and assayed for GFP enrichment at sites of dendrite-epidermis apposition. Whereas the single-pass transmembrane marker CD4-GFP broadly labeled epithelial membranes and showed no obvious enrichment at sites of dendrite contact (*Figure 1D and E*), our screen of ~90 GFP-tagged membrane- and cytoskeleton-associated proteins identified several markers enriched in basal domains of epithelial cells adjacent to c4da dendrites (*Figure 1—figure supplement 1A*, *Supplementary file 1*).

First, we screened a collection of membrane markers to determine whether ensheathment occurs at specialized membrane domains. Among these markers, the phosphatidylinositol 4,5-bisphosphate (PIP2) probe PLC$^\delta$-PH-GFP (*Várnai and Balla, 1998*; *Verstreken et al., 2009*) exhibited the most robust enrichment at sites of epidermal dendrite ensheathment. In epithelial cells of third instar larvae, PLC$^\delta$-PH-GFP accumulated at epithelial cell-cell junctions, punctate patches, and elongated filamentous membrane microdomains adjacent to c4da dendrites (*Figure 1F–1K*). These PIP2 microdomains were also labeled by antibodies to the *Drosophila* 4.1 protein cora (*Figure 1—figure supplement 1B*), a previously described marker of epidermal dendrite sheaths (*Kim et al., 2012*; *Tenenbaum et al., 2017*), demonstrating that these PIP2 microdomains correspond to epidermal dendrite sheaths. In addition to labeling epidermal sheaths, anti-cora immunostaining labels glial sheaths, which wrap axons, cell bodies, and proximal dendrite segments of sensory neurons; however, epidermal PLC$^\delta$-PH-GFP was not enriched at these sites of glial ensheathment. PLC$^\delta$-PH-GFP labeled membrane domains that often appeared wider than c4da dendrites (*Figure 1H–1I*), suggesting that PIP2 labels the entire sheath structure, including the convoluted tubular extensions to the basal surface of the epidermis. To more systematically analyze whether epidermal PIP2 microdomains mark sites of dendrite ensheathment by epidermal cells, we monitored staining intensity for a surface-exposed neuronal antigen (HRP) (*Kim et al., 2012*) and the epidermal PIP2 marker PLC$^\delta$-PH-GFP simultaneously. The intensity of HRP labeling along dendrites was inversely related to GFP labeling intensity, further suggesting that PLC$^\delta$-PH-GFP and hence PIP2 marks sites of neurite ensheathment (*Figure 1—figure supplement 2*).

Because many of the sheath structures are smaller than the axial resolution of a standard confocal microscope, we used expansion microscopy (ExM) to gain a 3-dimensional view of epidermal PLC$^\delta$-PH-GFP localization adjacent to c4da dendrites (*Jiang et al., 2018*). We found that PLC$^\delta$-PH-GFP labeled epidermal structures that extend from the most apical extent of dendrite insertion to the basal surface of individual epithelial cells (*Figure 1J–1K*), suggesting that PLC$^\delta$-PH-GFP indeed labels the entire sheath structure. PLC$^\delta$-PH-GFP was locally depleted at branch points (*Figure 1J*, white arrows; *Figure 1—figure supplement 1D*), consistent with prior observations that dendrite branch points are less extensively ensheathed than dendrite shafts (*Tenenbaum et al., 2017*). These sheath structures often appeared to terminate at epidermal cell-cell junctions, where dendrites were displaced to occupy domains basal to junctional domains. Point mutations in the PH domain of PLC$^\delta$-PH-GFP that abrogate PIP2 binding (*Várnai and Balla, 1998*; *Verstreken et al., 2009*) prevented accumulation of PLC$^\delta$-PH-GFP at sites of ensheathment (*Figure 1—figure supplement 1C*). Other PIP2-binding proteins, including OSH2-PH-GFP (*Figure 1—figure supplement 1E*), which binds phosphatidylinositol 4-phosphate and PIP2 with similar affinities (*Hardie et al., 2015*), exhibited similar patterns of accumulation at sheaths. Altogether, these observations demonstrate that epithelial sites of dendrite ensheathment are enriched in PIP2.

PIP2 is a negatively charged phospholipid that recruits a variety of proteins to the plasma membrane to regulate vesicular trafficking and actin remodeling (*De Craene et al., 2017*). We therefore examined whether endocytic, cytoskeletal, and/or phagocytic markers also accumulated at sites of epidermal ensheathment. Although we observed no enrichment of mature phagocytic markers prior to sheath formation or in mature sheaths, we identified a number of PIP2-linked markers that together provide a framework for sheath assembly (*Supplementary file 1*). First, we found that a GFP-tagged version of the endocytic adaptor Arf51F/dArf6 was enriched at sites of dendrite ensheathment (*Figure 1L–1M*). Arf6 regulates clathrin-dependent endocytosis as well as trafficking of recycling endosomes to the plasma membrane (*D'Souza-Schorey and Chavrier, 2006*), and the Arf6 effector phosphatidylinositol4-monophosphate 5-kinase catalyzes plasma membrane synthesis of PIP2 (*Honda et al., 1999*). Thus, dArf6 and endocytosis may contribute to PIP2 accumulation at sites of sheath formation. Second, we found that a GFP-tagged version of the GTPase Rho1, which promotes filamentous actin (F-actin) assembly, and the F-actin probe GMA-GFP accumulated at sites of epidermal sheath formation (*Figure 1N–1Q*), consistent with the fact that PIP2 stimulates actin assembly (*Yin and Janmey, 2003*). Finally, in addition to the septate junction marker cora (*Figure 1R–1S*), which was previously identified as a component of epidermal sheaths (*Kim et al., 2012*; *Tenenbaum et al., 2017*), other septate junction markers, including GFP-Neurexin-IV and Neuroglian-GFP, as well as adherens junction markers, including Armadillo-GFP and Shotgun-GFP, *Drosophila* homologues of β-catenin and E-cadherin, respectively, accumulated at epidermal dendrite sheaths (*Figure 1T–1U*, *Figure 1—figure supplement 1F*, *Supplementary file 1*). PIP2 binding regulates membrane association of 4.1R (*An et al., 2006*) and the maturation of adherens junctions via exocyst-dependent recruitment of E-cadherin (*Xiong et al., 2012*), thus PIP2 may promote sheath maturation via recruitment of these proteins.

## Epidermal sheaths are molecularly similar in the larval skin of *Drosophila* and zebrafish

Sensory axon terminals in the epidermis of zebrafish larvae and adults are ensheathed by the apical membranes of epidermal keratinocytes (*Figure 2A*) (*O'Brien et al., 2012*), and ensheathment channels have also been seen in adult fish (*Whitear and Moate, 1998*; *Rasmussen et al., 2018*). These axonal ensheathment channels are remarkably similar at the ultrastructural level to the sheaths wrapping somatosensory dendrites in *Drosophila* larvae. To determine whether zebrafish and *Drosophila* epidermal sheaths are similar at the molecular level, we examined the localization of fluorescent reporters for the membrane, cytoskeleton, and cell junctions in basal zebrafish epidermal cells.

At early stages, before sensory axons have grown into the skin, a reporter for PIP2 (PLC$^\delta$-PH-GFP) localized at cell-cell junctions and sparse microdomains near the apical surface (*Figure 2B*). After axons grew into the skin, PIP2 was enriched in continuous, linear apical microdomains, closely associated with axons of both larval zebrafish somatosensory neuron cell types, trigeminal and Rohon-Beard neurons (*Figure 2C,H*). Farnesylated GFP (CaaX-GFP) similarly localized to microdomains below axons, consistent with the notion that axons are associated with specialized membrane domains in skin cells (*Figure 2—figure supplement 1A–C*). Reporters for F-actin (LifeAct-GFP and

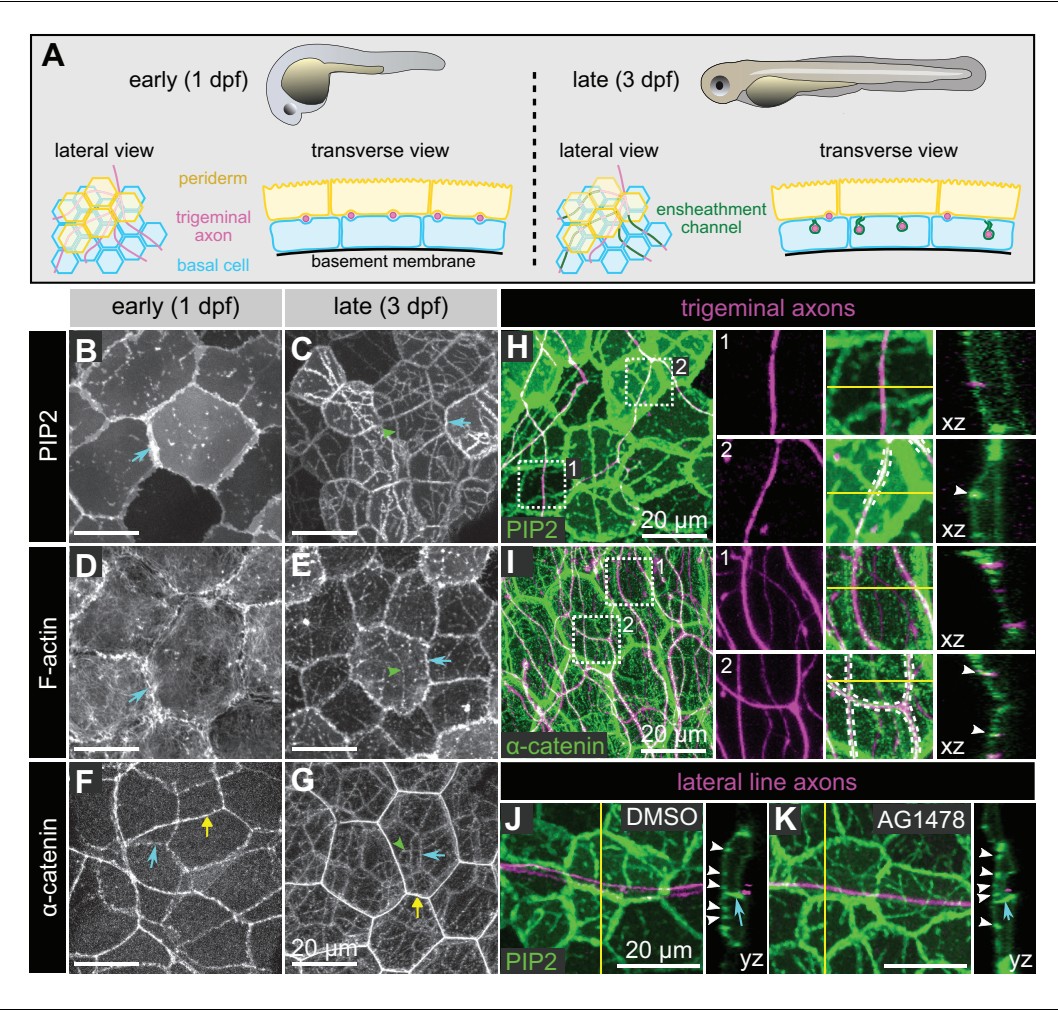

**Figure 2.** Molecular markers of epidermal sheaths in larval zebrafish. (A) Schematic of the bilayered larval zebrafish epidermis at the indicated stages based on the ultrastructural analysis (*O'Brien et al., 2012*). (B–G) Maximum intensity projections of confocal z-stacks showing lateral views through the epidermis at 24 hpf (B,D,F) or 72 hpf (C, E, G). Fluorescent reporters for PIP2 (B, C), F-actin (D, E), and α-catenin (F, G) are shown. Note the appearance of linear domains of each reporter through the apical basal cell membrane (green arrowheads) at the later time-point. Cyan arrows indicate basal lateral cell borders. Yellow arrows indicate periderm lateral cell borders. (H, I) Dual-labeling of epidermal sheaths and trigeminal sensory neurons. tdTomato-labeled trigeminal sensory neurons (magenta) together with the PIP2 reporter GFP-PH-PLC in basal cells at 46 hpf (H) or α-catenin-Citrine in both periderm and basal cells (I) at 73 hpf. Inset 1, examples of axons not associated with PIP2 (H) or α-catenin (I) enrichment. Inset 2, examples of axon-associated PIP2 (H) or α-catenin (I) enrichment. White dashed lines and arrowheads indicate examples of ensheathment channels containing labeled axons. Yellow lines indicate planes of orthogonal sections. (J, K) tdTomato-labeled posterior lateral line axons (magenta) labeled by transient injection of a *neurod:mTangerine* plasmid are shown together with GFP-PH-PLC signal in basal cells (green) at 78 hpf in either DMSO- or AG1478-treated embryos. AG1478 treatment prevents the repositioning of the posterior lateral line nerve below the epidermis (*Raphael et al., 2010*), resulting in the indentation of basal cell membranes, but did not trigger the accumulation of the PIP2 reporter GFP-PH-PLC. Arrowheads indicate ensheathment channels along the apical surface of basal cells. Cyan arrows indicate basal cell lateral borders. Yellow lines indicate planes of orthogonal sections. Note that because of the markers used, somatosensory and lateral line axons are sparsely labeled. Details of zebrafish experimental genotypes are provided in *Supplementary file 2*.
DOI: https://doi.org/10.7554/eLife.42455.007

The following figure supplement is available for figure 2:

**Figure supplement 1.** Membrane, adherens junction, and desmosome components localize to epidermal sheaths in zebrafish.
DOI: https://doi.org/10.7554/eLife.42455.008

Utrophin-GFP) were also enriched at these axon-associated domains (*Figure 2D and E* and data not shown).

Electron microscopy of zebrafish epidermal sheaths revealed that autotypic junctions appear to seal the 'neck' of these sheaths (*O'Brien et al., 2012*). To determine the molecular nature of these junctions, we used α-catenin and E-cadherin in-frame, functional gene traps (*Trinh et al., 2011*; *Cronan et al., 2018*); transiently expressed C-terminal-tagged Desmocolin-like two and Desmoplakin BAC reporters to visualize desmosomes; and a gene trap of Jupa [a.k.a. Plakoglobin/γ-catenin] (*Trinh et al., 2011*), a protein found in both types of junctions. Reporters for both adherens junction and desmosome proteins localized to apical domains directly above axons, suggesting that both types of junctions associate with epidermal sheaths (*Figure 2F–G,I*; *Figure 2—figure supplement 1D–O*). Consistent with the observation that autotypic junctions are only visible in some TEM images (*O'Brien et al., 2012*), some of the fluorescent junctional reporters (α-catenin, Dspa, Jupa) appeared as dotted lines along the length of axons (*Figure 2G*, *Figure 2—figure supplement 1J–O*), suggesting that they form spot junctions, rather than continuous belts.

Taken together, our results demonstrate similarity in ultrastructure and molecular composition of *Drosophila* and zebrafish epidermal sheaths, suggesting that these structures form via an evolutionarily conserved pathway.

## Ensheathment is specific to somatosensory neuron subtypes

To determine whether epidermal sheaths are specific to somatosensory neurons in zebrafish, or can occur at any site of axon-basal skin cell contact, we mislocalized axons of another sensory neuron type to the skin. Axons of posterior Lateral Line neurons (pLL) are usually separated from the skin by ensheathing Schwann cells, forming a nerve just internal to the epidermis. Treating animals with an inhibitor of the Neuregulin receptor Erbb3b, which is required for Schwann cell development, causes the entire bundle of pLL axons to directly contact the basal membrane of basal skin cells (*Raphael et al., 2010*). This treatment created a notable indentation in the basal membrane, but PLC$^\delta$-PH-GFP was not enriched in these domains (*Figure 2J and K*), indicating either that somatosensory axons can uniquely promote the formation of PIP2-rich microdomains, or that only the apical membranes of basal keratinocytes are competent to form these domains.

Next, we examined whether PIP2-rich microdomains formed around all somatosensory neurons or preferentially around particular subsets of somatosensory neurons. In *Drosophila* larvae, the vast majority of PIP2-positive sheath structures (94.8 ± 7.8%, n = 8 abdominal hemisegments) were present at sites occupied by sensory dendrites of da neurons (*Figure 3A–3C*). The few sheaths that were not apposed by dendrites were located directly adjacent to dendrites, suggesting that these sheaths may persist after ensheathed dendrites retracted. Next, to investigate whether different classes of da neurons were differentially ensheathed, we expressed membrane-targeted RFP in different classes of somatosensory neurons and visualized sheaths via epidermal expression of *UAS-PLC$^\delta$-PH-GFP* or anti-cora antibody staining. Among the multi-dendritic da neurons, we found that nociceptive c4da neurons exhibited the most extensive ensheathment, mechanosensitive and thermosensitive c3da and c2da neurons exhibited an intermediate level of ensheathment, and proprioceptive c1da neurons exhibited very little ensheathment (*Figure 3D–3F*, *Figure 3—figure supplement 1*). Thus, different morphological and functional classes of somatosensory neurons are ensheathed by the epidermis to different extents.

Although zebrafish somatosensory neurons have not been as clearly categorized into subtypes as *Drosophila* da neurons, similar to *Drosophila*, different individual sensory neurons in zebrafish were ensheathed to different degrees (*Figure 3G–3K*). The degree of ensheathment appeared to correlate with axon arbor complexity: axons with fewer branches associated with α-catenin along a greater proportion of their length (up to ~80% axon length) than did highly complex axons (<30% axon length). This observation implies that the degree of axon ensheathment may be a subtype-specific feature in zebrafish, like in *Drosophila*.

## Sheaths are not pre-patterned in the epidermis

As epidermal sheaths occur almost exclusively at sites occupied by sensory neurites, we investigated whether an epidermal pre-pattern dictates sites of sheath formation or, alternatively, whether neuronal signals induce epidermal sheath formation. To differentiate between these

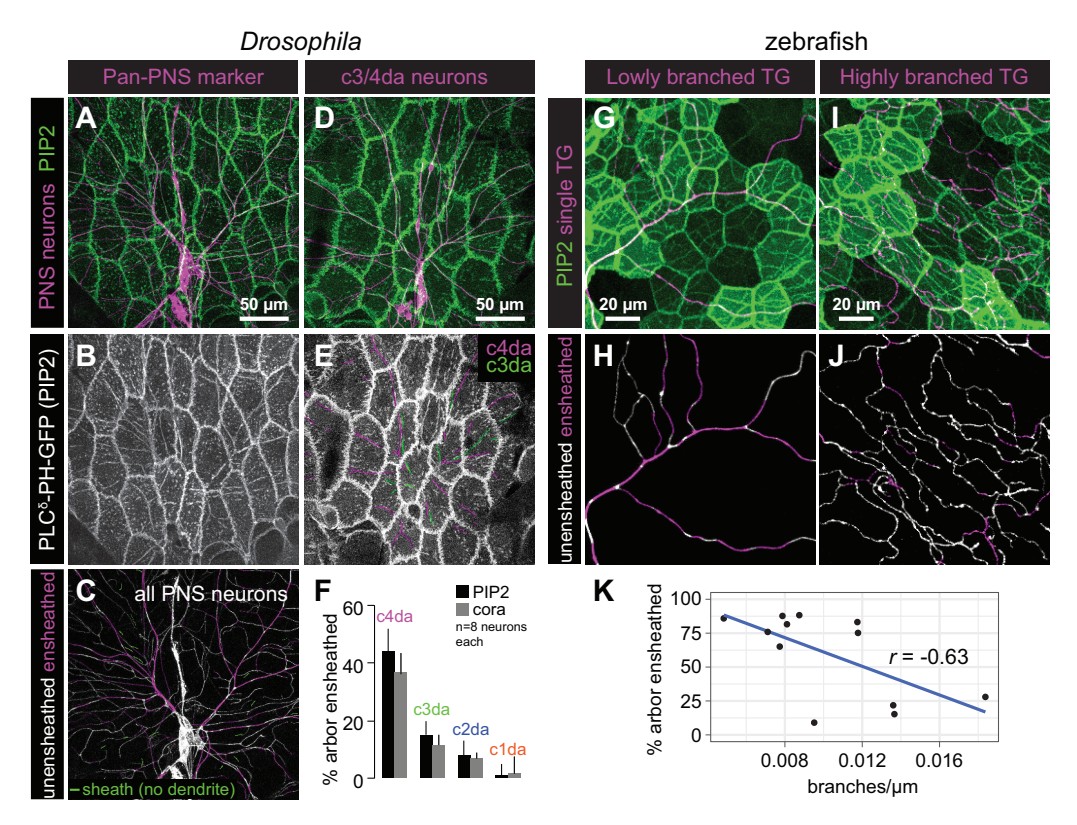

**Figure 3.** Epithelial sheaths form adjacent to somatosensory neurons in a modality-specific manner. (A–C) Dual-labelling of epithelial sheaths and all somatosensory neurons. Maximum projections of confocal stacks show (A) mRFP-labeled sensory neurons (magenta) together with epidermal PLC$^\delta$-PH-GFP signal (green) or (B) epidermal PLC$^\delta$-PH-GFP signal on its own. (C) PLC$^\delta$-PH-GFP-positive sheaths are pseudocolored with dendrite-associated sheaths shaded in magenta and sheaths without apposed dendrites shaded in green. (D, E) Dual-labeling of epithelial sheaths and c3da/c4da sensory neurons. (D) tdTomato-labeled c3da and c4da neurons (magenta) are shown together with epidermal PLC$^\delta$-PH-GFP signal (green). (E) Image showing epidermal PLC$^\delta$-PH-GFP signal with c3da-containing sheaths shaded green and c4da-containing sheaths shaded magenta. (F) Histogram depicting mean and standard deviation values for the portion of the dendrite arbor of different classes of da neurons ensheathed by the epidermis using PLC$^\delta$-PH-GFP or cora immunostaining as a marker for ensheathment. (G–K) The extent of ensheathment was inversely correlated with trigeminal (TG) axon complexity in zebrafish. Examples of single TG neurons labeled by transient injection of *Tg(isl1[ss]:LEXA-VP16,LEXAop:tdTomato)* with low (G, H) or high (I, J) branch density. Epidermal sheaths are shaded in magenta in (H, J). (K) Scatterplot of axon branches versus percentage of axon length ensheathed from tracings of 12 individual TG neurons. Note the inverse linear regression (blue line).

DOI: https://doi.org/10.7554/eLife.42455.009

The following figure supplement is available for figure 3:

**Figure supplement 1.** Dual-labeling of epithelial sheaths and c1da (top) or c2da (bottom neurons).

DOI: https://doi.org/10.7554/eLife.42455.010

possibilities, we first monitored the timing of arrival and distribution of epidermal sheath markers throughout *Drosophila* larval development. Whereas c4da dendrites tile the larval body wall by ~36 h after egg laying (AEL) (*Parrish et al., 2009*), PIP2 first accumulated in isolated patches adjacent to dendrites at 48 h AEL (*Figure 4A–4C and G*). Epidermal PIP2 did not co-occur with large portions of the dendrite arbor until after 96 hAEL (*Figure 4D–4F and G*), a time point at which dendrites are internalized in epithelial cells (*Jiang et al., 2014*; *Jiang et al., 2018*). Furthermore, time-lapse imaging demonstrated that PIP2 enrichment at sheaths is not transient; once formed, PIP2-positive epidermal sheaths persist or grow, but rarely retract (*Figure 4H*, *Figure 4—figure supplement 1*). Finally, although PIP2 markers and cora extensively co-localized and labeled a nearly identical population of sheaths by the end of larval development (95.7 ± 5.8% of cora-positive sheaths are PIP2-positive; 88.7 ± 7.4% of PIP2-positive sheaths are cora-positive; n = 8 hemisegments), cora accumulation lagged behind PLC$^\delta$-PH-GFP (*Figure 4G*, *Figure 4—figure supplement 2*). Thus, although

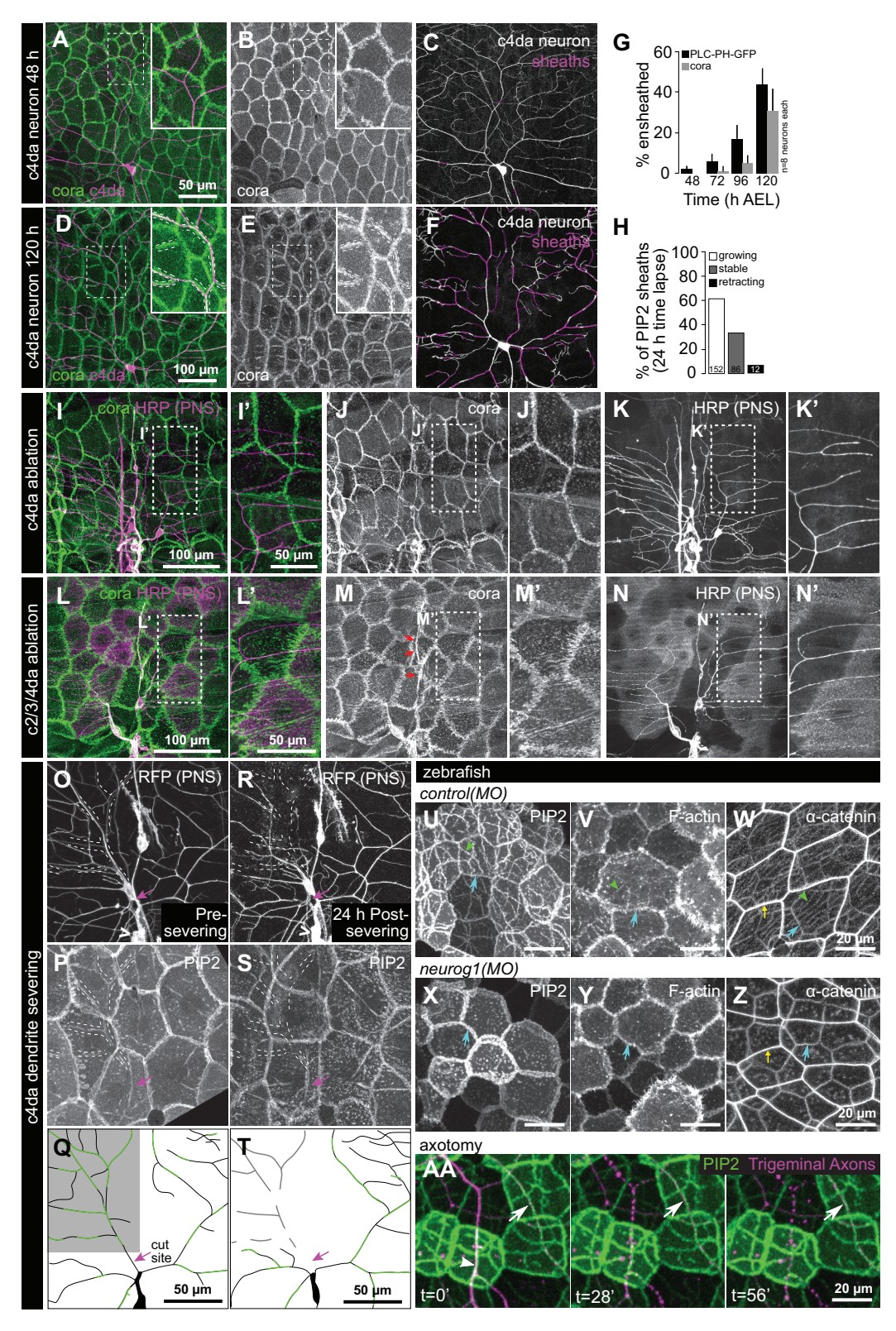

**Figure 4.** Somatosensory neurons are necessary for formation and maintenance of epidermal sheaths. (A–G) Timecourse of sheath formation. Maximum intensity projections show dual labeling of sheaths by epidermal PLC$^\delta$-PH-GFP and the c4da-specific marker ppk-CD4-tdTomato (A, D) or PLC$^\delta$-PH-GFP signal alone (B, E) at 48 and 120 h AEL. (C, F) Composites show portions of c4da dendrite arbors (shaded purple) wrapped by sheaths labeled by PLC$^\delta$-PH-GFP. (G) Plot shows mean and standard deviation values for the proportion of c4da dendrite arbors wrapped by PLC$^\delta$-PH-GFP or cora-positive
*Figure 4 continued on next page*

*Figure 4 continued*

sheaths at the indicated time points. See also *Figure 4—figure supplement 2* for images of cora labeling of sheaths at 72 and 120 h AEL. (**H**) Once formed, sheaths persist. Plot shows sheath dynamics; the proportion of sheaths from eight neurons that grew, retracted, or were stable over a 24 h time-lapse is shown. See also *Figure 4—figure supplement 1* for time-lapse images. (**I–K**) Epidermal sheath formation following genetic ablation of c4da neurons. Maximum intensity projections show dual labeling of anti-cora staining to label sheaths and anti-HRP staining to label PNS neurons (**I**) and the individual markers alone (**J, K**) at 120 h AEL for a larva expressing the pro-apoptotic gene *reaper* (*rpr*) specifically in c4da neurons under control of *ppk-Gal4*. (**L–N**) Epidermal sheath formation following laser ablation of larval c2da, c3da, and c4da neurons. Images show dual labeling of epidermal sheaths with anti-cora staining and sensory neurons with anti-HRP staining (**L**) and the individual markers alone (**M, N**) at 120 h AEL in a hemisegment in which c2da, c3da, and c4da were ablated with a focused laser beam at 72 h AEL. Red arrowheads in (**M**) mark cora-positive structures (also visible in *Figure 1R* and *Figure 4—figure supplement 2*) located in the dorsal-medial portion of the hemisegment, corresponding to the position of the dorsal pharyngeal sense organ. These structures are not co-labeled by other epidermal sheath markers and cora immunoreactivity persists following epidermal *cora(RNAi)* (*Figure 5—figure supplement 2*), suggesting that these structures are distinct from the epidermal sheaths that wrap sensory neurites. (**O–T**) Somatosensory dendrites are required for sheath maintenance. Maximum projections of confocal stacks show time-lapse images of da neurons labeled with membrane-targeted mRFP (**O**) and epidermal sheaths (**P**) immediately prior to c4da dendrite severing at 108 h AEL and 12 h post-severing at 120 h AEL (**R, S**). White dashed lines outline the anterior-dorsal portions of the c4da arbor that are ensheathed prior to severing and the location those sheaths would occupy if they persisted post-severing. (**Q, T**) Traces depict unensheathed c4da dendrites in black and ensheathed c4da dendrites in green, the arrow marks the site of dendrite severing, and the gray box marks the quadrant in which c4da dendrites and associated epidermal sheaths are lost post-severing. (**U–Z**) Epidermal sheath formation in zebrafish injected with a morpholino targeting *neurog1* to prevent somatosensory neuron development. Maximum intensity projections of confocal z-stacks showing lateral views through the zebrafish epidermis at 72 hpf. Note the lack of ensheathment channels (green arrowheads) in *neurog1(MO)*-injected embryos. Yellow and cyan arrows indicate the lateral cell membranes of periderm and basal cells, respectively. (**AA**) Somatosensory axons are required for sheath maintenance in zebrafish. Arrowhead and arrow indicate sheaths associated with a severed and intact axon, respectively.

DOI: https://doi.org/10.7554/eLife.42455.011

The following figure supplements are available for figure 4:

**Figure supplement 1.** Time-lapse imaging of epidermal sheaths.

DOI: https://doi.org/10.7554/eLife.42455.012

**Figure supplement 2.** Absence of an epidermal cora pre-pattern prior to epidermal sheath formation.

DOI: https://doi.org/10.7554/eLife.42455.013

**Figure supplement 3.** Epidermal PIP2 microdomains sparsely form in hemisegments lacking c4da neurons.

DOI: https://doi.org/10.7554/eLife.42455.014

PIP2 accumulation marks an earlier stage in sheath formation than does cora recruitment, we found no evidence that a pre-pattern predicts the site of ensheathment.

In the course of our imaging we occasionally observed hemisegments lacking a c4da neuron. In such cases, epidermal PIP2-positive sheath structures were largely absent, although PIP2 accumulation at epithelial cell-cell junctions was comparable to neighboring segments containing c4da neurons (*Figure 4—figure supplement 3*). This observation suggested that dendritic signals induce formation of epidermal sheaths.

## Peripheral sensory neurites are required for sheath formation and maintenance

To test the requirement for sensory neurons in epidermal sheath formation, we used a genetic cell-killing assay in *Drosophila* to eliminate all c4da neurons and assayed for sheath formation using anti-cora immunostaining. Expressing the pro-apoptotic gene *reaper* (*rpr*) in c4da neurons with two copies of the c4da-specific *ppk-Gal4* Gal4 driver (*Grueber et al., 2003*) resulted in fully penetrant death and clearance of c4da neurons but not other sensory neurons by the end of the first larval instar, prior to appearance of epithelial sheaths. Anti-cora staining of these larvae revealed that although the overall extent of ensheathment was significantly reduced, levels of ensheathment in c1/c2/c3da neurons were unaffected by this treatment (cora-positive sheath length per $mm^2$ of body wall: $2.72 \pm 0.64$ mm following c4da *rpr* expression; $11.44 \pm 1.81$ mm in sibling controls without *rpr*; $3.18 \pm 1.16$ mm for c1/c2/c3da neurons from sibling controls; mean $\pm$sd, n = 8) (*Figure 4I–4K*). These results demonstrate that dendrite-derived signals induce sheath formation; such signals are likely short-range signals, as sheaths form at sites directly apposed to dendrites. These results further suggest that modality-specific levels of ensheathment do not reflect competitive interactions between c4da and other da neurons for sheath formation, as the absence of c4da neurons did not potentiate sheath formation in spared neurons.

Next, we investigated the temporal requirement for dendrite-derived signals in epidermal sheath formation. Using a focused laser beam we ablated *Drosophila* c2da, c3da, and c4da neurons at 48 h AEL, prior to appreciable accumulation of sheath markers or appearance of sheaths in TEM sections (*Jiang et al., 2014*), and assayed for sheath formation at 120 h AEL using anti-cora immunostaining. Following this treatment, cora-positive sheaths did not form (*Figure 4L–4N*), suggesting that dendrite signals initiate sheath formation after 48 h AEL, the same timeframe at which PIP2 markers first accumulate at sites of dendrite contact. These results further demonstrate that different neuron classes have different capacities for ensheathment, because removing all of the da neurons that are normally ensheathed did not potentiate c1da neuron ensheathment.

To examine whether dendritic signals are likewise required for sheath maintenance, we used a focused laser beam to sever the dorsal-anterior dendrites from a c4da neuron at 108 h AEL, after epidermal sheaths had formed, and used time-lapse confocal microscopy to monitor effects on sheath maintenance in larvae expressing the sheath marker *UAS-PLC$^\delta$-PH-GFP* (*Figure 4O–4Q*). By 12 h post-severing, both the c4da dendrites distal to the cut site and the epidermal sheaths that wrapped them had disappeared (*Figure 4R–4T*). By contrast, sheaths wrapping the spared dorsal-posterior portion of the c4da dendrite arbor, as well as sheaths that wrapped c2da/c3da neurons in both the lesioned and unlesioned half of the hemisegment persisted. Therefore, short-range dendrite-derived signals are required both for the formation and maintenance of epidermal sheaths.

To determine whether, as in *Drosophila*, axons are required for formation of epidermal sheaths in zebrafish, we examined sheath-associated reporters in larvae injected with a morpholino targeting *neurogenin 1* (*neurog1*), a manipulation that blocks somatosensory neuron development (*Andermann et al., 2002*; *Cornell and Eisen, 2002*; *O'Brien et al., 2012*). Basal cells in *neurog1* MO-treated animals lacked coherent PIP2-rich microdomains, apical accumulations of F-actin, and α-catenin-containing autotypic junctions, demonstrating that epidermal sheaths are initiated by axons in zebrafish larvae (*Figure 4U–4Z*). As in *Drosophila*, axons were also required to maintain sheaths, as PIP2-rich microdomains disappeared soon after laser axotomy and axon degeneration (*Figure 4AA*).

## Zebrafish axonal sheaths and *Drosophila* dendritic sheaths form in a similar sequence

To determine the order of assembly of these sheath-associated proteins, we conducted a series of double-labeling and genetic epistasis analyses in *Drosophila* larvae. We simultaneously expressed the PIP2 marker *UAS-PLC$^\delta$-PH-Cerulean* together with either the endocytic marker *UAS-dArf6-GFP* or the F-actin marker *UAS-GMA-GFP* in the epidermis of larvae additionally expressing the c4da neuron marker *ppk-CD4-tdTomato* and monitored the timing of arrival of each marker at epidermal sheaths. From the earliest time-point that PIP2 enrichment was detectable at sheaths, we also detected dArf6-GFP enrichment, albeit at a subset of PIP2-positive sheaths, suggesting that dArf6 is recruited to sheaths shortly after PIP2 enrichment (*Figure 5A–5B*). By contrast, F-actin labeling lagged behind PIP2 (*Figure 5C–5D*), appearing on a comparable timescale as cora.

To directly visualize the stepwise recruitment of sheath components, we labeled epidermal sheaths with the PIP2 marker *UAS-PLC$^\delta$-PH-Cerulean* and assayed for recruitment of GFP-tagged sheath components using time-lapse microscopy. Consistent with our time-lapse imaging of PIP2-positive sheaths (*Figure 4—figure supplement 1*), we found that PLC$^\delta$-PH-Cerulean labeling persisted at the vast majority of sheaths over a 12 h time-lapse (*Figure 5E–5F*). However, sheaths that were initially labeled by *UAS-PLC$^\delta$-PH-Cerulean* but not *UAS-GFP-cora$^{1-383}$*, a GFP-tagged fusion protein that mimics endogenous cora localization at epidermal sheaths and septate junctions (*Figure 5—figure supplement 1C*), were positive for both markers following a 12 h time-lapse (*Figure 5E–5F*). Similarly, we found that GMA-GFP and dArf6-GFP were recruited to sheaths that were initially PIP2-positive but GFP-negative (*Figure 5F*, *Figure 5—figure supplement 1A–1B*). Epidermal sheath assembly therefore appears to proceed via separable steps.

Examining ensheathment channel-associated markers at four stages of zebrafish development revealed a similar sequence of events. As in *Drosophila*, we found that membrane reporters appeared near zebrafish axons before F-actin or junctional reporters (*Figure 5G*). PIP2-rich microdomains frequently apposed axons by 32 hpf, before ensheathment channels were evident ultrastructurally (*O'Brien et al., 2012*). This observation suggests that the formation of PIP2-positive membrane microdomains is an early step in sheath morphogenesis in zebrafish, as in *Drosophila*.

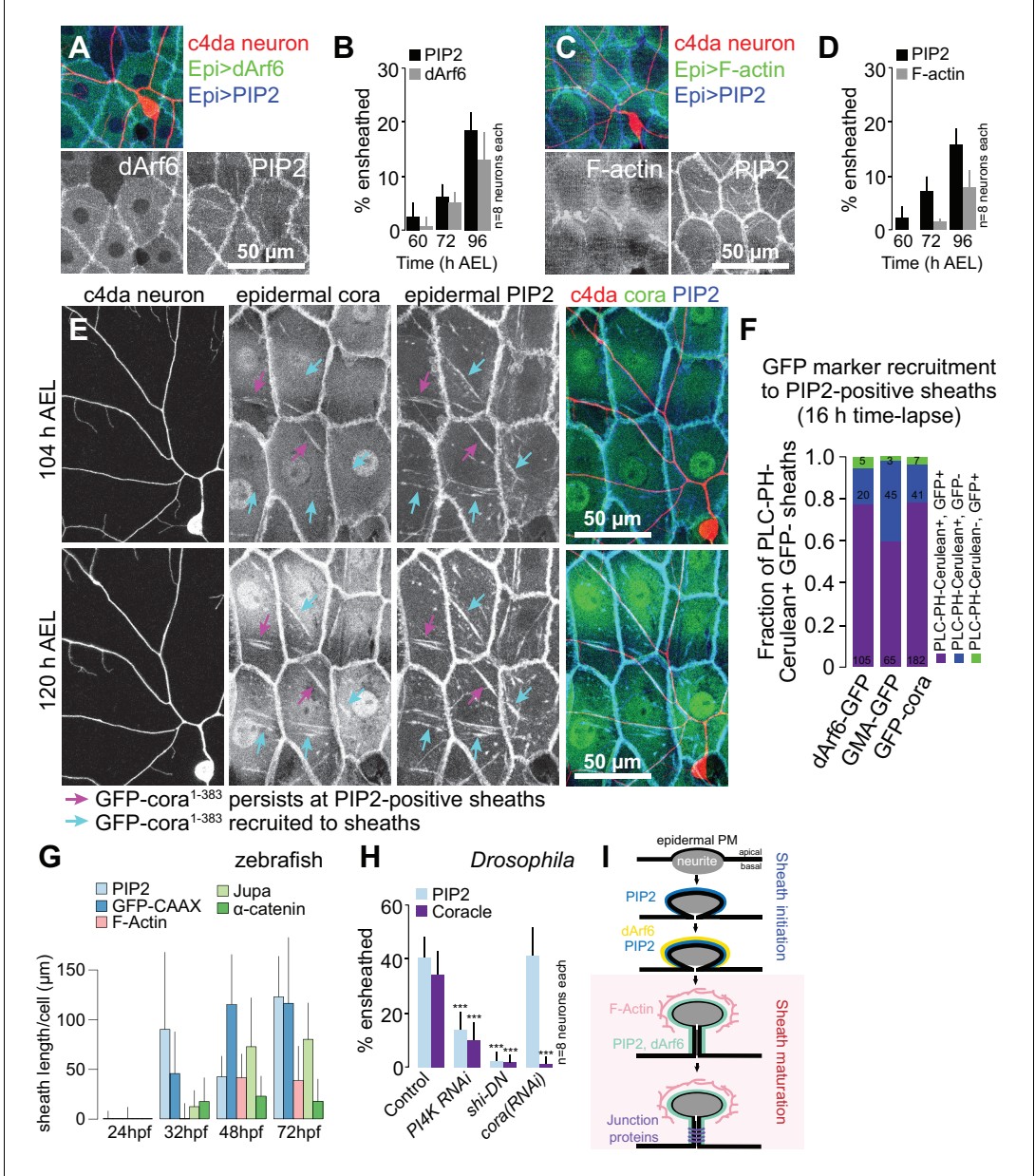

**Figure 5.** Sequence of events in sheath assembly. (**A–D**) Time of arrival of PIP2 and other sheath markers. Images show dual labeling of sheaths by PLC$^\delta$-PH-Cerulean and dArf6-GFP (**A**) or GMA-GFP to label F-actin (**C**) in larvae additionally expressing the c4da-specific marker ppk-CD4-tdTomato. (**B, D**) Plots show mean and standard deviation values for the proportion of c4da dendrite arbors ensheathed by structures labeled by the indicated markers at the indicated times. All sheath structures labeled by dArf6-GFP and GMA-GFP were labeled by PLC$^\delta$-PH-Cerulean. (**E**) Time-lapse images show dual labeling of sheaths by PLC$^\delta$-PH-Cerulean and GFP-cora$^{1\text{-}383}$ in larvae additionally expressing the c4da-specific marker ppk-CD4-tdTomato. Cyan arrows mark single-positive (Cerulean-positive) sheaths that convert to double-positive (Cerulean-positive and GFP-positive), magenta arrows mark double-positive sheaths that persist. See also *Figure 5—figure supplement 1* for time-lapse images depicting dArf6-GFP and GMA-GFP recruitment to PLC$^\delta$-PH-Cerulean sheaths. (**F**) Quantification of marker recruitment. Bars depict the proportion of PLC$^\delta$-PH-Cerulean-positive GFP-negative sheaths at 104 h AEL that are labeled by the indicated markers at 120 h AEL. More than 100 sheaths (from six independent time-lapse series) were scored for each marker combination. (**G**) Timing of accumulation of ensheathment channel markers in the zebrafish epidermis. (**H**) Epistatic relationship between markers. The indicated RNAi transgenes were expressed in the epidermis and effects on ensheathment were assessed (see *Figure 5—figure supplement 2* for accompanying images). Plots show mean and standard deviation values for the proportion of c4da dendrite arbors wrapped by PLC$^\delta$-PH-GFP or cora-positive sheaths. n = 8 neurons each; ***p<0.001 relative to control; one way ANOVA with post-hoc Dunnett's test. (**I**) Model depicting the timing of arrival of sheath components.

DOI: https://doi.org/10.7554/eLife.42455.015

The following figure supplements are available for figure 5:

*Figure 5 continued on next page*

*Figure 5 continued*

**Figure supplement 1.** Sequential recruitment of epidermal sheath proteins.
DOI: https://doi.org/10.7554/eLife.42455.016
**Figure supplement 2.** PIP2 microdomains form shortly after axons innervate the zebrafish epidermis.
DOI: https://doi.org/10.7554/eLife.42455.017
**Figure supplement 3.** Epistasis analysis of sheath-associated proteins.
DOI: https://doi.org/10.7554/eLife.42455.018

Indeed, time-lapse confocal microscopy demonstrated that these domains formed during development just minutes after an axonal grown cone passed through that region (*Figure 5—figure supplement 2*).

To assess the relationship between these sheath-associated proteins, we knocked down lipids or proteins associated with sheaths in *Drosophila*. Specifically, to deplete phosphatidylinositol 4-phosphate and PIP2, we expressed RNAi targeting the phosphatidylinositol 4-kinase gene *PI4KIIIα*; to block endocytosis, we expressed a dominant negative version of *shibire* (*shi^DN*), which is defective in GTP binding/hydrolysis (*Damke et al., 2001*); to block septate junction formation, we expressed *cora(RNAi)* in the epidermis. We found that epidermal *PI4K(RNAi)* and *shi^DN* expression severely attenuated PIP2 accumulation at sheaths (*Figure 5H*, *Figure 5—figure supplement 3*). As PLC$^δ$-PH-GFP accumulation precedes dArf6 accumulation at the onset of sheath formation, PIP2 accumulation and endocytic events may engage in feed-forward signaling to promote epidermal sheath formation. By contrast, epidermal *cora(RNAi)* had no effect on PLC$^δ$-PH-GFP accumulation, suggesting that cora accumulation is a downstream event in sheath assembly. Consistent with this notion, both epidermal *PI4K(RNAi)* and *shi^DN* expression blocked cora accumulation at sheaths (*Figure 5H*, *Figure 5—figure supplement 3*), suggesting that cora recruitment to sheaths depends on PIP2 accumulation. PIP2 accumulation and cora accumulation therefore mark genetically separable steps in sheath assembly that we subsequently refer to as initiation and maturation, respectively (*Figure 5I*).

## Epidermal sheaths regulate dendrite growth dynamics and structural plasticity

What are the functions of epidermal sheaths that wrap somatosensory neurons? Prior studies suggested a role for epidermal ensheathment in restricting dendrite branching in *Drosophila* larvae (*Jiang et al., 2014*; *Tenenbaum et al., 2017*). We therefore assayed the requirement in dendrite growth of each of the sheath assembly components we identified in this study. We expressed *PI4K (RNAi)* to reduce epidermal PIP2 levels and monitored effects on c4da dendrite morphogenesis. Compared to controls, epidermis-specific expression of *PI4K(RNAi)* significantly increased the number and decreased the average length of terminal dendrites (*Figure 6A–6B and G–H*). PLC$^δ$-PH-GFP can function as a competitive inhibitor of PIP2 signaling (*Raucher et al., 2000*), and epidermal PLC$^δ$-PH-GFP expression increased terminal dendrite branch number and decreased dendrite branch length in a dose-dependent manner (*Figure 6—figure supplement 1*). Similarly, blocking epidermal endocytosis via constitutive epidermal expression of *shi^DN* or expressing temperature sensitive *shi^ts* and using it to conditionally blocking epidermal endocytosis specifically in the time window during which dendrites are normally ensheathed led to severe terminal dendrite branching defects qualitatively similar to *PI4K(RNAi)* (*Figure 6C–6D and G–H*). Finally, epidermal expression of *cora(RNAi)* induced growth of short terminal dendrites (*Figure 6E and G–H*), as has been previously reported (*Tenenbaum et al., 2017*), as did epidermal expression of *shg(RNAi)* (*Figure 6F–6H*). Thus, blocking the early or late events of epidermal sheath formation deregulates branching morphogenesis of *Drosophila* nociceptive c4da neurons.

To identify the cellular basis of these dendrite growth defects, we monitored dendrite dynamics in control or sheath-defective larvae using time-lapse microscopy during the time window when sheaths normally form. Over an 18 h time-lapse beginning at 96 h AEL more than 80% of terminal dendrites persisted in control larvae, with the vast majority of those dendrites elongating (*Figure 6I and L*). By contrast, using epidermis-specific expression of *PI4K(RNAi)* or *cora(RNAi)* to block sheath initiation or maturation, respectively, led to significant alterations in branch dynamics (*Figure 6J–6L*). First, a larger fraction of terminal dendrites exhibited dynamic growth behavior. Second, the

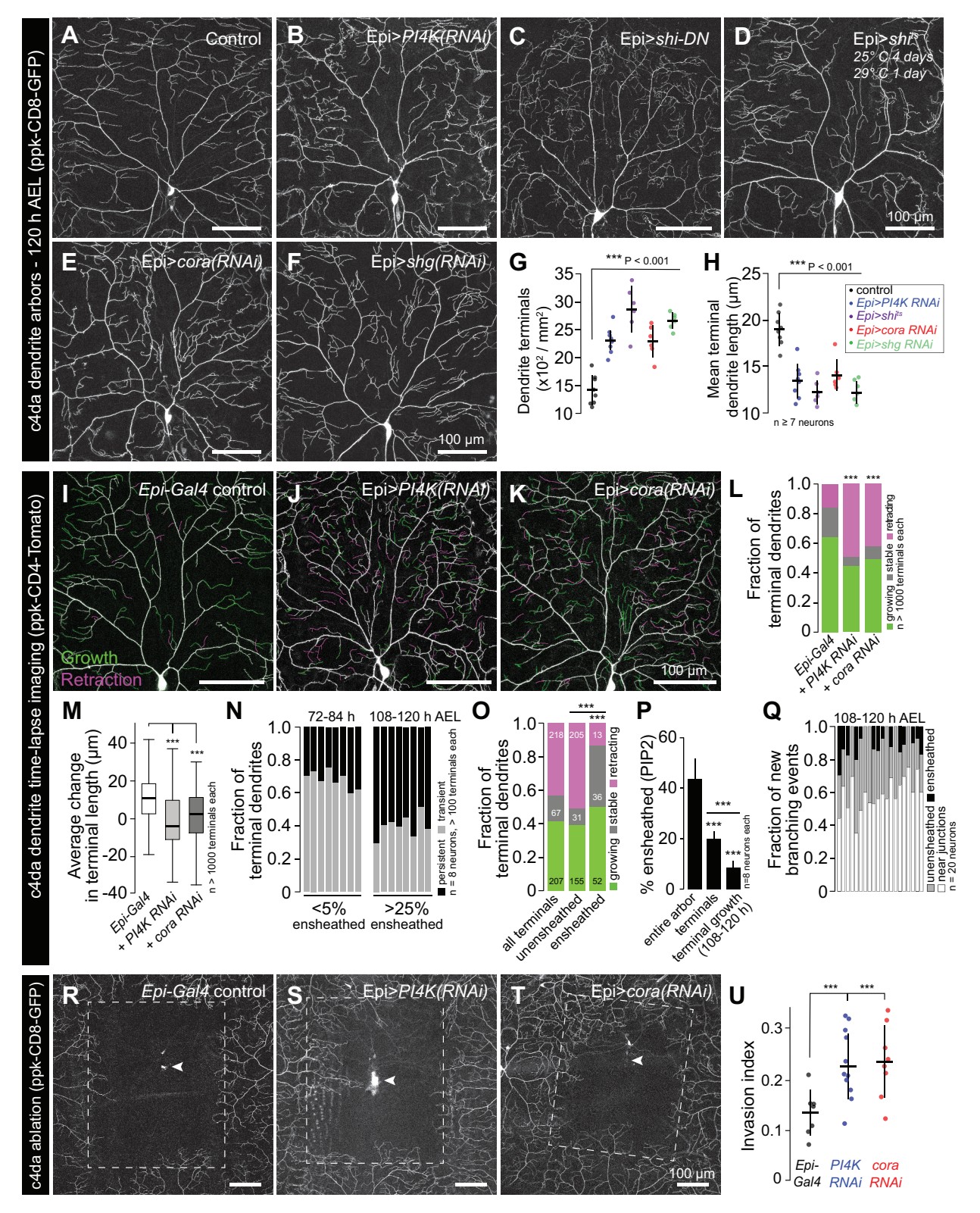

**Figure 6.** Epidermal sheaths regulate branching morphogenesis in nociceptive c4da neurons. Representative images of 120 h AEL c4da neurons from (A) control larvae and larvae expressing (B) *PI4K(RNAi)*, (C) dominant-negative *shibire (shi^DN^)*, (D) temperature-sensitive *shibire (shi^ts^)*, (E) epidermal *cora (RNAi)*, and (F) epidermal *shg(RNAi)* larvae are shown. Larvae were reared at 25° C, with the exception of larvae in (D) which were reared at 25° C for 4 days and then shifted to the non-permissive temperature 29° C for 1 day prior to imaging. (G–H) Morphometric analysis of dendrites from c4da neurons

*Figure 6 continued on next page*

*Figure 6 continued*

of the indicated genotypes. Plots show mean and standard deviation for (**G**) the number of terminal branches and (**H**) terminal branch length. Data points, measurements from an individual neuron; ***p<0.001 relative to control; one way ANOVA with post-hoc Dunnett's test. (**I–L**) Time-lapse analysis of epidermal sheath control of terminal dendrite dynamics. C4da neurons were imaged over an 18 h time-lapse (96–114 h AEL) and growth (green) and retraction (magenta) were pseudocolored in a composite of the two time-points. Representative composite images are shown for c4da neurons from (**I**) *Gal4*-only control, (**J**) epidermal *PI4K(RNAi)*, and (**K**) epidermal *cora(RNAi)* larvae. (**L–P**) Quantification of terminal dendrite dynamics. (**L**) The fraction of terminal dendrites that were growing, stable, or retracting over the time-lapse is shown. ***p<0.001 compared to controls, Chi-square analysis. (**M**) Epidermal ensheathment regulates the extent of terminal dendrite dynamics. Box plots depict mean values and 1st/3rd quartile, whiskers mark minimum/maximum values. ***p<0.001 compared to Epi-Gal4 control; one way ANOVA with post-hoc Dunnett's test. (**N**) Epidermal ensheathment regulates dendrite turnover. C4da neurons were imaged over a 12 h time-lapse (72–84 or 108–120 h AEL) and all terminal dendrites were scored as persistent (present at both time points) or transient. Each bar represents measurements from a single neuron. Terminal dendrites at the later time-point, when c4da neurons are extensively ensheathed, were significantly more likely to persist. (**O**) Quantification of terminal dynamics in ensheathed and unensheathed terminal dendrites from 108 to 120 h AEL. ***p<0.001, Chi-square analysis with post-hoc Bonferroni adjustment for multiple comparisons. Pairwise comparisons are indicated. (**P**) Dynamic portions of dendrite arbors are less extensively ensheathed. Mean and standard deviation values for the proportion of c4da dendrite arbors, terminal dendrites, and new terminal dendrite growth (12 h time-lapse) wrapped by PLC$^\delta$-PH-GFP sheaths at 120 h AEL. ***p<0.001, Chi-square analysis with post-hoc Bonferroni adjustment for multiple comparisons. Pairwise comparisons are indicated. (**Q**) Distribution of branching events during 12 h time-lapse imaging. Each bar represents a single neuron. (**R–U**) Epidermal ensheathment regulates dendrite structural plasticity. Class IV neurons in newly eclosed 2nd instar control (**R**), epidermis *PI4k(RNAi)* (**S**), and epidermis *cora(RNAi)* (**T**) larvae were ablated with a focused laser beam and imaged 48 h post-ablation. Images depict dendrite growth of spared neurons into unoccupied territory following laser ablation and hatched boxes demarcate the territory occupied by the ablated neuron. (**U**) Scatter plot depicting mean and standard deviation for dendrite invasion of the indicated mutants. The number of samples analyzed for each treatment is indicated. ***p<0.001 relative to control; one way ANOVA with post-hoc Dunnett's test.

DOI: https://doi.org/10.7554/eLife.42455.019

The following figure supplements are available for figure 6:

**Figure supplement 1.** Epidermal PLC$^\delta$-PH-GFP dosage affects c4da dendrite development.

DOI: https://doi.org/10.7554/eLife.42455.020

**Figure supplement 2.** Proximal-distal distribution of epidermal sheaths.

DOI: https://doi.org/10.7554/eLife.42455.021

**Figure supplement 3.** Time-lapse analysis of ensheathment at dynamic portions of dendrite arbors.

DOI: https://doi.org/10.7554/eLife.42455.022

relative levels of growth and retraction were altered; whereas growth predominated in controls, growth and retraction occurred with comparable frequency in *PI4K(RNAi)* and *cora(RNAi)* larvae. Third, the average change in terminal dendrite length was reduced in *PI4K(RNAi)* and *cora(RNAi)* larvae (*Figure 6M*).

These results suggest that epidermal ensheathment alters dendrite growth properties by stabilizing existing terminal dendrites and promoting their elongation. To further test this possibility, we simultaneously labeled epidermal sheaths (*Epi > PLC$^\delta$-PH-GFP*) and c4da dendrite arbors (*ppk-CD4-tdTomato*) and monitored terminal dendrite dynamics in ensheathed and unensheathed arbors. Whereas >65% of terminal dendrites were present only transiently during a 12 h time lapse at the onset of ensheathment (72–84 h AEL), most terminal dendrites persisted during a 12 h time lapse after arbors were extensively ensheathed (108–120 h AEL) (*Figure 6N*). In this latter time window (108–120 h AEL) we compared the growth dynamics of ensheathed and unensheathed terminal dendrites and found that a significantly higher proportion of ensheathed terminal dendrites were growing or stable over the 12 h time-lapse (*Figure 6O*). Altogether, our time-lapse imaging results strongly suggest that epidermal sheaths contribute to stabilization of somatosensory dendrites.

What is the relationship between epidermal ensheathment and dendrite branching? While dendrite branch points are occasionally ensheathed (*Figure 1B*) and new branches can be initiated from ensheathed dendrites (*Han et al., 2012*), we found that sheath formation is first initiated on long-lived dendrite shafts in proximal portions of the dendrite arbor rather than the more dynamic distal portions of the dendrite arbor (*Figure 6—figure supplement 2*), and that terminal dendrites in general and newly formed terminal dendrites in particular were less extensively ensheathed than other portions of the dendrite arbor (*Figure 6P*, *Figure 6—figure supplement 3*). We therefore monitored the frequency of dendrite branching from ensheathed and unensheathed portions of dendrite arbors during a 12 h time-lapse. Consistent with prior observations (*Han et al., 2012*), we occasionally observed new branch initiation from ensheathed portions of dendrite arbors (*Figure 6Q*).

However, these events were rare and usually occurred at the ends of existing sheaths (*Figure 6—figure supplement 3*); the majority of new branch initiation occurred on unensheathed portions of dendrites. Intriguingly, a large proportion of new branches was formed in the vicinity of epithelial intercellular junctions; whether this is simply a result of discontinuities in sheaths at intercellular junctions or reflects the function of non-autonomous branch-promoting activities associated with junctions remains to be determined.

Given that epidermal ensheathment constrains terminal dendrite dynamics in *Drosophila*, we next examined whether epidermal ensheathment limits structural plasticity of dendrite arbors, as has been suggested (*Parrish et al., 2009*; *Jiang et al., 2014*). Embryonic ablation of c4da neurons leads to exuberant dendrite growth in spared neurons beyond their normal boundaries to fill vacated territory (*Grueber et al., 2003*; *Sugimura et al., 2003*). This capacity of c4da neurons to expand their dendrite arbors beyond normal boundaries is progressively limited during development, concomitant with the increase in epidermal dendrite ensheathment (*Parrish et al., 2009*; *Jiang et al., 2014*). Following ablation of a single c4da neuron at 72 h AEL, the spared neighboring neurons extend their dendrite arbors to cover 13% of the vacated territory, on average (*Figure 6R and U*). If epithelial ensheathment limits the structural plasticity of c4da dendrite arbors, we reasoned that blocking epithelial sheath formation should potentiate the invasive growth activity of c4da neurons following ablation of their neighbors. Indeed, epidermis-specific *PI4K(RNAi)* or *cora(RNAi)* resulted in a significant potentiation of dendrite invasion (*Figure 6S and U*). In addition to regulating the growth dynamics and elongation of individual terminal dendrites, these results suggest that epidermal ensheathment contributes to the fidelity of receptive field coverage by coupling dendrite and epidermis expansion.

## Epidermal sheaths regulate nociception in *Drosophila* larvae

What role, if any, does epidermal ensheathment play in somatosensation? Having found that nociceptive c4da neurons and proprioceptive c1da neurons were the most extensively and least ensheathed da neurons, respectively, we investigated whether blocking sheath formation affected sensory-evoked behavioral responses regulated by these neurons. Harsh touch activates c4da nociceptive neurons to elicit stereotyped nocifensive rolling responses (*Zhong et al., 2010*), so we monitored touch-evoked rolling responses and rates of larval locomotion in control or sheath-defective larvae as a measure for sheath influence on c4da neuron function. Stimulation with a 78 nM von Frey filament induced nociceptive rolling behavior in >60% of control larvae, whereas c4da-specific expression of the inward rectifying potassium channel Kir2.1 strongly attenuated this rolling response (*Figure 7A*). Compared to controls, epidermal expression of either *PI4KIIIa(RNAi)* to block PIP2 accumulation or *PIS(RNAi)* to reduce phophoinositol biosynthesis, or feeding larvae the cell permeant polyphosphoinositide-binding peptide PBP10 to antagonize PIP2 signaling during the time window of sheath formation significantly attenuated mechanonociceptive behavior (*Figure 7A*, *Figure 7—figure supplement 1*). Epidermal expression of *shi$^{DN}$* to block epidermal endocytosis and *cora(RNAi)* to block sheath maturation similarly attenuated mechanonociception. We additionally found that previously reported treatments that block ensheathment including overexpressing α- and β-integrin in c4da neurons to tether dendrites to the ECM (*Han et al., 2012*; *Jiang et al., 2014*) and mutation of the miRNA *bantam* (*Jiang et al., 2014*) displayed reduced rolling rates in response to von Frey stimuli.

Finally, we assayed for effects of ensheathment on larval locomotion. Input from proprioceptive c1da neurons is required for coordinated larval locomotion, and perturbing c1da neuron function severely attenuates larval crawling speed (*Song et al., 2007*). Treatments that reduced epidermal sheath formation did not reduce larval stride length or crawling speed as would be expected for disruption of proprioceptor function, but instead led to increased larval crawling speed (*Figure 7B* and data not shown). This increased crawling speed was largely the result of reduced turning frequency and a concomitant increase in forward-directed locomotion (*Figure 7C*), similar to defects in crawling trajectory induced by perturbing c4da function (*Ainsley et al., 2003*; *Gorczyca et al., 2014*), further suggesting that ensheathment modulates c4da function. Thus, epidermal ensheathment potentiates nociceptive mechanosensory responses and is apparently dispensable for proprioceptor function, consistent with our observation that nociceptive c4da but not proprioceptive c1da neurons exhibit extensive epidermal ensheathment.

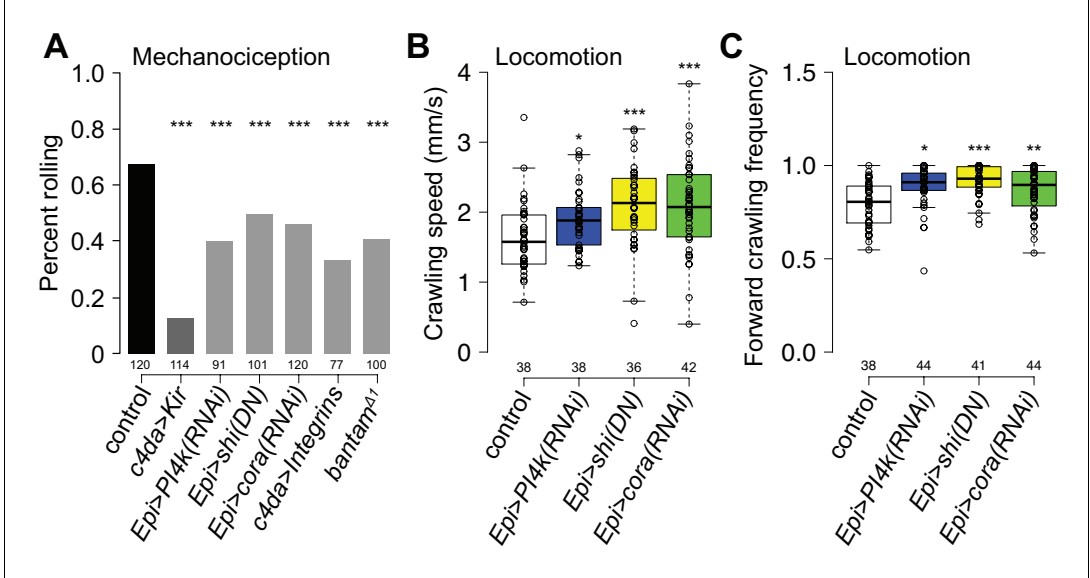

**Figure 7.** Epidermal dendrite ensheathment regulates nociceptive sensitivity. (**A**) Bars depict the proportion of larvae of the indicated genotype that exhibited a nocifensive rolling response to 70 mN von Frey fiber stimulation. *UAS-Kir2.1* expression in nociceptive c4da neurons blocked nociceptive responses to 70 mN stimulus, demonstrating that the response is mediated by c4da neurons, and treatments that reduced epidermal ensheathment significantly reduced the frequency of nociceptive rolling responses. \*\*\*p<0.001, compared to wt controls, Chi square test. (**B, C**) Box plots depict crawling speed (**B**) and the proportion of time larvae spent in forward-directed locomotion (**C**) for larvae of the indicated genotype. \*\*\*p<0.001, \*p<0.05, Kruskal-Wallis rank sum test. The number of larvae tested is shown for each condition.

DOI: https://doi.org/10.7554/eLife.42455.023

The following figure supplement is available for figure 7:

**Figure supplement 1.** PBP10 feeding inhibits sheath formation.

DOI: https://doi.org/10.7554/eLife.42455.024

## Discussion

A neuron's function is profoundly influenced by its interaction with cells around it. In the skin, specialized interactions with epidermal cells influence the function of a variety of different sensory neurons. However, despite the fact that keratinocytes are the most abundant cell type in the epidermis, roles for keratinocyte-sensory neuron interactions in somatosensation are still not well characterized. Here, we have identified a conserved morphogenetic program for ensheathment of peripheral somatosensory neurites by epidermal cells. In both *Drosophila* and zebrafish, sensory neurite-derived signals induce epidermal cells to ensheath somatosensory neurons in a neuron type-specific manner. These neurite-derived signals induce local formation of epidermal PIP2-enriched membrane microdomains that are essential for ensheathment, local assembly of F-actin, and recruitment of junctional proteins that likely seal the sheaths.

### What triggers formation of epidermal sheaths?

Although the signals are not yet known, our studies define key features of the signaling system that drives sheath formation. First, epidermal sheath formation likely relies on short-range, contact-mediated signals involving neuron-expressed ligands and epidermal receptors, as sheaths form exclusively at sites occupied by peripheral sensory neurites. Such a signaling system bears similarity to the *C. elegans* epidermal SAX-7/L1CAM and MNR-1/Menorin co-ligand complex that interacts with neuronal DMA-1 to regulate patterning of PVD dendrites (*Dong et al., 2013*; *Salzberg et al., 2013*). However, whereas PVD dendrites are positioned according to a hypodermal grid of SAX-7/L1CAM expression (*Liang et al., 2015*), the location of epidermal sheaths is dependent on neuron-derived signals rather than an epidermal pre-pattern. Second, different types of neurons have different capacities to induce epidermal sheath formation; in zebrafish, only somatosensory neurons are capable of inducing sheath formation on the apical membranes of basal keratinocytes, and different

classes of somatosensory neurons are ensheathed to different degrees in *Drosophila* and zebrafish. The epidermal sheaths that wrap different types of somatosensory neurons are structurally similar, thus it seems likely that different levels of the sheath-inducing ligand determine the extent of ensheathment much as Nrg1 levels can drive the extent of Schwann cell ensheathment (*Michailov et al., 2004*). Based on the conservation in the molecular machinery of sheath formation, such a ligand and its epidermal receptor are likely conserved in fish and flies. Third, sheath formation is temporally regulated. In both *Drosophila* and zebrafish, somatosensory neurites innervate the epidermis more than a day prior to sheath formation (*Parrish et al., 2009*; *O'Brien et al., 2012*). This may reflect a lack of competence by epithelial cells to ensheath somatosensory neurites as accelerating developmental progression in the *Drosophila* epidermis leads to precocious dendrite ensheathment (*Jiang et al., 2014*). Finally, our laser severing experiments suggest that peripheral neurites are required to maintain epidermal sheaths. Whether maintenance of sheaths is dependent on a dedicated maintenance signal or simply reflects the absence of morphogenetic signals that would remodel sheaths, for example the exposure by neurites to phosphatidylserine or other engulfment-promoting signals, remains to be determined. Epidermally embedded dendrites that lack identifiable sheath-like structures have been previously described (*Han et al., 2012*); whether such structures represent cases in which sheaths have been lost or form via a distinct developmental mechanism remains to be determined.

The earliest epidermal morphogenetic event we identified downstream of neurite-derived ensheathment signals is the appearance of PIP2-enriched membrane microdomains. How might neurite-derived signals trigger local accumulation of epidermal PIP2? Two prominent mechanisms exist to form localized pools of PIP2 in the plasma membrane (*Kwiatkowska, 2010*), and each can be triggered by cell-cell contacts. First, PIP2 can be locally clustered via electrostatic interactions with polybasic proteins such as myristoylated alanine-rich C-kinase substrate (MARCKS) (*Glaser et al., 1996*; *Gambhir et al., 2004*; *McLaughlin and Murray, 2005*), which additionally binds and cross-links filamentous actin (*Myat et al., 1997*). Protocadherins regulate cortical dendrite morphogenesis in part by maintaining a membrane-associated pool of active MARCKS (*Garrett et al., 2012*), thus protocadherin-based adhesion provides one potential mechanism for localizing MARCKS and hence PIP2 in epidermal cells. Neuronal signals could likewise trigger PIP2 localization via engagement of transmembrane receptors with intracellular domains that electrostatically interact with and cluster PIP2 (*McLaughlin and Murray, 2005*) or via membrane recruitment of other polybasic proteins such as adducins or GAP43 (*Kwiatkowska, 2010*). Second, PIP2 can be locally synthesized, most commonly via phosphorylation of phosphatidylinositol 4-phosphate, and type I phosphatidylinositol 4-phosphate five kinase (PIP5KI) can associate with N-cadherin to locally produce PIP2 at sites of N-cadherin adhesion (*El Sayegh et al., 2007*). PIP5KIγ associates with the exocyst via direct interaction with Exo70 to promote membrane targeting of E-cadherin (*Xiong et al., 2012*), thus cadherin-based adhesion can be both a cause and effect of localized PIP2 synthesis. Although we have not found evidence for an epidermal PIP2 pre-pattern that determines sites of ensheathment, PIP5K additionally localizes to focal adhesions to provide a local source of PIP2 (*Ling et al., 2002*). Thus, it will be intriguing to determine whether integrin-based adhesions contribute to epidermal sheath formation by generating local asymmetries in PIP2 levels that get amplified by neuron-derived signals.

Plasma membrane enrichment of epidermal PIP2 serves as a critical control point for a variety of cellular processes (*Sun et al., 2013*). Among these, we note remarkable similarities between epidermal sheath formation and the early events of phagocytosis. First, sheath formation and the early stages of phagocytosis appear to involve similar cellular rearrangements, with ensheathing cells and engulfing cells wrapping their targets with membrane protrusions. Second, sheath formation and phagocytosis share a common set of molecular mediators as PIP2 accumulates in nascent epidermal sheaths and in the phagocytic cup of engulfing cells (*Botelho et al., 2000*), as does a network of F-actin (*Scott et al., 2005*). Third, many types of ensheathing cells additionally exhibit phagocytic activity, including *Drosophila* and zebrafish keratinocytes (*Han et al., 2014*; *Rasmussen et al., 2015*), *Drosophila* ensheathing glia (*Doherty et al., 2009*), and astrocytes (*Chung et al., 2013*). However, whereas PIP2 levels persist at sheaths, PIP2 disappears from the phagosomal membrane during the late stages of phagocytosis (*Botelho et al., 2000*), leading to disassembly of the associated actin network (*Scott et al., 2005*). Similarly, transient accumulation of PIP2 is a feature of endocytosis, cell migration, and other PIP2 regulated morphogenetic events. Thus, reducing PIP2 levels

may facilitate phagocytic engulfment of neurites, providing a mechanism for rapid conversion of the epidermal ensheathment channels to engulfment channels.

## Functional roles for epidermal neurite ensheathment

Consistent with prior reports, we found that epidermal ensheathment limits dendrite branching of *Drosophila* nociceptive c4da neurons (*Jiang et al., 2014*; *Tenenbaum et al., 2017*). We also found that the extent of ensheathment is inversely related to peripheral axon branch number in zebrafish somatosensory neurons, suggesting that epidermal ensheathment could similarly regulate neurite branching in vertebrates. This epidermal growth control of peripheral sensory arbors appears to involve two related mechanisms. First, epidermal ensheathment limits dendrite branching; dendrite branching events rarely occur on ensheathed dendrites, and blocking epidermal ensheathment potentiates dendrite branching. This dendrite branching control may reflect a masking of dendrite arbors from substrate-derived signals that promote branching or a steric hindrance of branching. Second, epidermal ensheathment stabilizes existing neurites; blocking epidermal ensheathment potentiates dynamic growth behavior and structural plasticity in *Drosophila* sensory neurons. Determining whether ensheathment similarly regulates structural plasticity in zebrafish will require development of more and better tools for effectively blocking sheath formation in zebrafish. However, given that the timing of epidermal sheath formation correlates with the developmental restriction in structural plasticity in both *Drosophila* and zebrafish (*O'Brien et al., 2012*; *Jiang et al., 2014*), developmental control of ensheathment appears to be a likely mechanism to stabilize receptive fields of somatosensory neurons.

Different types of somatosensory neurons appear to be ensheathed to different degrees. What would be the purpose of such an arrangement? Many different types of somatosensory neurons innervate overlapping territories, and one recent study suggests that selective ensheathment of particular sensory neuron types facilitates coexistence of different types of sensory neurons in a given territory (*Tenenbaum et al., 2017*). Differential levels of ensheathment may additionally allow for differential coupling of somatosensory neurons to epidermal growth-promoting signals. Likewise, differential ensheathment of somatosensory neuron types may allow different levels of functional coupling of sensory neurons and epidermis. Our finding that nociceptive c4da neurons are the most extensively ensheathed *Drosophila* somatosensory neurons, and that ensheathment regulates nociceptive sensitivity, suggests that epidermal ensheathment may play a particularly important role in tuning responses to noxious stimuli. Intriguingly, mutations that block ensheathment impair the function of a subset of *C. elegans* mechanosensory neurons (*Chen and Chalfie, 2014*); whether these mechanosensory impairments are a consequence of ensheathment defects or other effects of the mutations remains to be determined.

How might epidermal sheaths influence nociceptive sensitivity? First, epidermal sheaths may potentiate the functional coupling of epidermal cells to somatosensory neurons. Recent studies suggest that sensory-evoked responses of keratinocytes may modulate sensory neuron function (*Koizumi et al., 2004*; *Baumbauer et al., 2015*; *Pang et al., 2015*; *Moehring et al., 2018*), and epidermal sheaths could provide sites for vesicle release from keratinocytes or direct electrical coupling between keratinocytes and somatosensory neurons. Merkel cells provide a precedent for the former possibility (*Maksimovic et al., 2013*), but whether keratinocytes possess presynaptic release machinery and which neurotransmitters they express remain to be determined. Alternatively, epidermal ensheathment could potentiate nociceptor sensitivity by increasing proximity to stimulus source, by clustering sensory channels, or by some other means. Regardless of the mechanism, our findings that epidermal ensheathment modulates nociceptive sensitivity suggest that defects in epidermal ensheathment could contribute to sensory deficits in human disease. Intriguingly, some forms of peripheral neuropathy exhibit loss of unmyelinated intraepidermal nerves (*Weis et al., 2011*; *Üçeyler et al., 2013*); whether defects in epithelial ensheathment play a role in these sensory neuropathies remains to be determined.

## Materials and methods

**Key resources table**

*Continued on next page*

*Continued*

| Reagent type | Designation | Source or reference | Identifiers | Additional information |
|---|---|---|---|---|
| Reagent type | Designation | Source or reference | Identifiers | Additional information |
| Gene (*D. melanogaster*) | *bantam* | NA | FLYB:FBgn0262451 | |
| Gene (*D. melanogaster*) | *coracle* | NA | FLYB:FBgn0010434 | |
| Gene (*D. melanogaster*) | *dArf6* | | | FlyBase symbol: Arf51F |
| Gene (*D. melanogaster*) | *Pis* | NA | FLYB:FBgn0030670 | |
| Gene (*D. melanogaster*) | *PI4KII* | NA | FLYB:FBgn0037339 | FlyBase symbol: Pi4KIIα |
| Gene (*D. melanogaster*) | *shg* | NA | FLYB:FBgn0003391 | |
| Gene (*D. melanogaster*) | *shi* | NA | FLYB:FBgn0003392 | |
| Genetic reagent (*D. melanogaster*) | *w1118* | Bloomington Drosophila Stock Center | BDSC:5905; FLYB:FBal0018186 | Parrish Lab stock |
| Genetic reagent (*D. melanogaster*) | *ppk-CD4-tdTomato* | Bloomington Drosophila Stock Center | BDSC:35844; FLYB:FBti0143430 | FlyBase symbol: P{ppk-CD4-td Tomato}4a |
| Genetic reagent (*D. melanogaster*) | *ppk-CD4-tdTomato* | Bloomington Drosophila Stock Center | BDSC:35845; FLYB:FBti0143432 | FlyBase symbol: P{ppk-CD4-td Tomato}10a |
| Genetic reagent (*D. melanogaster*) | *Epidermal-Gal4* | PMID: 15269788 | FLYB:FBti0072310 | FlyBase symbol: P{GAL4}A58 |
| Genetic reagent (*D. melanogaster*) | *elav-LexA* | Bloomington Drosophila Stock Center | BDSC:52676; FLYB:FBti0155565 | FlyBase symbol: P{GMR27E08-lexA}attP40 |
| Genetic reagent (*D. melanogaster*) | *lexAOP-CD4-tdTomato* | Bloomington Drosophila Stock Center | BDSC:77138; FLYB:FBti0195760 | FlyBase symbol: P{13xLexAop2-CD4-tdTom}4 |
| Genetic reagent (*D. melanogaster*) | *NompC-LexA* | Bloomington Drosophila Stock Center | BDSC:52240; FLYB:FBti0157008 | FlyBase symbol: PBac{nompC-lexA::p65}VK00018 |
| Genetic reagent (*D. melanogaster*) | *ppk-Gal4* | Bloomington Drosophila Stock Center | BDSC:32078; FLYB:FBti0127690 | FlyBase symbol: P{ppk-GAL4.G}2 |
| Genetic reagent (*D. melanogaster*) | *ppk-Gal4* | Bloomington Drosophila Stock Center | BDSC:32079; FLYB:FBti0131208 | FlyBase symbol: P{ppk-GAL4.G}3 |
| Genetic reagent (*D. melanogaster*) | *21–7 Gal4* | PMID: 20696376 | FLYB:FBti0157010 | FlyBase symbol: P{GAL4}21–7 |
| Genetic reagent (*D. melanogaster*) | *98b-Gal4* | PMID: 26063572 | FLYB:FBti0169386 | FlyBase symbol: P{GAL4}98b |
| Genetic reagent (*D. melanogaster*) | *GMR37B02-Gal4* | PMID: 20697123 | FLYB: FBti0135266 | FlyBase symbol: P{GMR37B02-GAL4}attP2 |
| Genetic reagent (*D. melanogaster*) | *UAS-rpr* | Bloomington Drosophila Stock Center | BDSC:5824; FLYB:FBti0016094 | FlyBase symbol: P{UAS-rpr.C}14 |
| Genetic reagent (*D. melanogaster*) | *UAS-PIS(RNAi)* | Bloomington Drosophila Stock Center | BDSC:29383; FLYB:FBti0129011 | FlyBase symbol: P{TRiP.JF03315}attP2 |
| Genetic reagent (*D. melanogaster*) | *UAS-PI4KII(RNAi)* | Bloomington Drosophila Stock Center | BDSC:38242; FLYB:FBti0144268 | FlyBase symbol: P{TRiP.GL00179}attP2 |

*Continued on next page*

*Continued*

| Reagent type | Designation | Source or reference | Identifiers | Additional information |
|---|---|---|---|---|
| Genetic reagent (*D. melanogaster*) | *UAS-cora(RNAi)* | Bloomington Drosophila Stock Center | BDSC:51845; FLYB:FBti0157811 | FlyBase symbol: P{TRiP.HMC03418}attP40 |
| Genetic reagent (*D. melanogaster*) | *UAS-shg(RNAi)* | Bloomington Drosophila Stock Center | BDSC:32904; FLYB:FBti0140407 | FlyBase symbol: P{TRiP.HMS00693}attP2 |
| Genetic reagent (*D. melanogaster*) | *UAS-shits* | Bloomington Drosophila Stock Center | BDSC:44222; FLYB:FBti0151794 | FlyBase symbol: P{UAS-shits1.K}3 |
| Genetic reagent (*D. melanogaster*) | *UAS-shiDN* | Bloomington Drosophila Stock Center | BDSC:5811; FLYB:FBti0016096 | FlyBase symbol: P{UAS-shi.K44A}4–1 |
| Genetic reagent (*D. melanogaster*) | *UAS-mys* | PMID: 10572057 | FLYB:FBal0102506 | FlyBase symbol: mysUAS.cDa |
| Genetic reagent (*D. melanogaster*) | *UAS-mew* | PMID: 9250662 | FLYB:FBal0062567 | FlyBase symbol: mewUAS.cMBa |
| Genetic reagent (*D. melanogaster*) | *UAS-Kir2.1-GFP* | Bloomington Drosophila Stock Center | BDSC:6596; FLYB:FBti0017551 | FlyBase symbol: P{UAS-Hsap\KCNJ2.EGFP}1 |
| Genetic reagent (*D. melanogaster*) | *Cha$^{7.4kb}$-Gal80* | PMID: 19531155 | FLYB:FBtp0089195 | FlyBase symbol: P{ChAT.7.4kb-Gal80} |
| Genetic reagent (*D. melanogaster*) | *bantamΔ1* | PMID: 12196398 | FLYB:FBab0029992 | FlyBase symbol: Df(3L)banΔ1 |
| Genetic reagent (*D. melanogaster*) | *UAS-CD4-tdGFP* | Bloomington Drosophila Stock Center | BDSC:35836; FLYB:FBti0143423 | FlyBase symbol: PBac{UAS-CD4-tdGFP}VK00033 |
| Genetic reagent (*D. melanogaster*) | *UAS-mCD8-GFP* | Bloomington Drosophila Stock Center | BDSC:5137; FLYB:FBti0012685 | FlyBase symbol: P{UAS-mCD8::GFP.L}LL5 |
| Genetic reagent (*D. melanogaster*) | *UAS-myr-GFP* | Bloomington Drosophila Stock Center | BDSC:32198; FLYB:FBti0131964 | FlyBase symbol: P{10XUAS-IVS-myr::GFP}attP40 |
| Genetic reagent (*D. melanogaster*) | *UAS-Asap-GFP* | Bloomington Drosophila Stock Center | BDSC:65849; FLYB:FBti0183078 | FlyBase symbol: P{UASp-Asap.GFP}attP2 |
| Genetic reagent (*D. melanogaster*) | *UAS-Dl-GFP* | Bloomington Drosophila Stock Center | BDSC:8610; FLYB:FBti0058796 | FlyBase symbol: P{UAS-Dl::GFP}DA53 |
| Genetic reagent (*D. melanogaster*) | *UAS-CG10702-GFP* | Bloomington Drosophila Stock Center | BDSC:65857; FLYB:FBti0183086 | FlyBase symbol: P{UASp-CG10702.GFP}attP2 |
| Genetic reagent (*D. melanogaster*) | *UAS-Dlg5-GFP* | Bloomington Drosophila Stock Center | BDSC:30928; FLYB:FBti0130055 | FlyBase symbol: P{UAS-Dlg5.GFP}2 |
| Genetic reagent (*D. melanogaster*) | *Sdc-CPTI002578* | Kyoto Drosophila Genomics and Genetics Resources | Kyoto:115306; FLYB:FBal0261862 | FlyBase symbol: SdcCPTI002578 |
| Genetic reagent (*D. melanogaster*) | *UAS-2xOsh2PH-GFP* | Bloomington Drosophila Stock Center | BDSC:57353; FLYB:FBti0162453 | FlyBase symbol: P{UAS-2xOsh2PH-GFP}attP2 |
| Genetic reagent (*D. melanogaster*) | *UAS-PLCδ-PH-GFP* | Bloomington Drosophila Stock Center | BDSC:39693; FLYB:FBti0148832 | FlyBase symbol: P{UAS-PLCδ-PH-EGFP}3 |
| Genetic reagent (*D. melanogaster*) | *UAS-PLCD1-PH-Cerulean* | Bloomington Drosophila Stock Center | BDSC:30895; FLYB:FBti0129991 | FlyBase symbol: P{UASp-PLCD1PH.Cerulean}2 |

*Continued on next page*

*Continued*

| Reagent type | Designation | Source or reference | Identifiers | Additional information |
|---|---|---|---|---|
| Genetic reagent (*D. melanogaster*) | *UAS-PLCδ-PH-GFP (S39R)* | Bloomington Drosophila Stock Center | BDSC:39694; FLYB:FBti0148833 | FlyBase symbol: P{UAS-PLCδ-PH.S39R-EGFP}3 |
| Genetic reagent (*D. melanogaster*) | *UAS-step-GFP* | Bloomington Drosophila Stock Center | BDSC:65862; FLYB:FBti0183091 | FlyBase symbol: P{UASp-step.EGFP}attP2 |
| Genetic reagent (*D. melanogaster*) | *UAS-GFP-myc-2xFYVE* | Bloomington Drosophila Stock Center | BDSC:42712; FLYB:FBti0147756 | FlyBase symbol: P{UAS-GFP-myc-2xFYVE}2 |
| Genetic reagent (*D. melanogaster*) | *UAS-Arf51F-GFP* | Bloomington Drosophila Stock Center | BDSC:65867; FLYB:FBti0183094 | FlyBase symbol: P{UASp-Arf51F.GFP}attP2 |
| Genetic reagent (*D. melanogaster*) | *UAS-Arf79F-GFP* | Bloomington Drosophila Stock Center | BDSC:65850; FLYB:FBti0183079 | FlyBase symbol: P{UASp-Arf79F.GFP}attP3 |
| Genetic reagent (*D. melanogaster*) | *UAS-Arf102F-GFP* | Bloomington Drosophila Stock Center | BDSC:65866; FLYB:FBti0183075 | FlyBase symbol: P{UASp-Arf102F.GFP}attP4 |
| Genetic reagent (*D. melanogaster*) | *UAS-Arl4-GFP* | Bloomington Drosophila Stock Center | BDSC:65868; FLYB:FBti0183092 | FlyBase symbol: P{UASp-Arl4.GFP}attP5 |
| Genetic reagent (*D. melanogaster*) | *UAS-Clc-GFP* | Bloomington Drosophila Stock Center | BDSC:7109; FLYB:FBti0027887 | FlyBase symbol: P{UAS-EGFP-Clc}5 |
| Genetic reagent (*D. melanogaster*) | *UAS-gamma-cop-GFP* | Bloomington Drosophila Stock Center | BDSC:29711; FLYB:FBti0128586 | FlyBase symbol: P{UASp-γCOP.EGFP}3 |
| Genetic reagent (*D. melanogaster*) | *UAS-YFP-Rab4* | Bloomington Drosophila Stock Center | BDSC:9767; FLYB:FBti0100783 | FlyBase symbol: P{UASp-YFP.Rab4}Sap-r32 |
| Genetic reagent (*D. melanogaster*) | *UAS-YFP-Rab-5* | Bloomington Drosophila Stock Center | BDSC:24616; FLYB:FBti0100788 | FlyBase symbol: P{UASp-YFP.Rab5}02 |
| Genetic reagent (*D. melanogaster*) | *UAS-GFP-Rab-7* | Bloomington Drosophila Stock Center | BDSC:42705; FLYB:FBti0150335 | FlyBase symbol: P{UASp-YFP.Rab7}2 |
| Genetic reagent (*D. melanogaster*) | *UAS-YFP-Rab11* | Bloomington Drosophila Stock Center | BDSC:50782; FLYB:FBti0152903 | FlyBase symbol: P{UASp-YFP.Rab11}3 |
| Genetic reagent (*D. melanogaster*) | *UAS-YFP-Rab-21* | Bloomington Drosophila Stock Center | BDSC:23242; FLYB:FBti0100844 | FlyBase symbol: P{UAST-YFP.Rab21}smog04 |
| Genetic reagent (*D. melanogaster*) | *UAS-YFP-Rab35* | Bloomington Drosophila Stock Center | BDSC:9821; FLYB:FBti0100873 | FlyBase symbol: P{UASp-YFP.Rab35}15 |
| Genetic reagent (*D. melanogaster*) | *UAS-spin-RFP* | Bloomington Drosophila Stock Center | BDSC:42716; FLYB:FBti0147756 | FlyBase symbol: P{UAS-GFP-myc-2xFYVE}2 |
| Genetic reagent (*D. melanogaster*) | *UAS-Shrub-GFP* | Bloomington Drosophila Stock Center | BDSC:32559; FLYB:FBti0131611 | FlyBase symbol: P{UAS-shrb-GFP}2 |
| Genetic reagent (*D. melanogaster*) | *UAS-dia-GFP* | Bloomington Drosophila Stock Center | BDSC:56751; FLYB:FBti0161167 | FlyBase symbol: P{UASp-dia.EGFP}2 |

*Continued*

| Reagent type | Designation | Source or reference | Identifiers | Additional information |
|---|---|---|---|---|
| Genetic reagent (*D. melanogaster*) | *UAS-EndoA-GFP* | This paper | | Created by injecting *pJFRC7-EndoA-GFP* DNA into flies with the *attP2* docking site |
| Genetic reagent (*D. melanogaster*) | *UAS-EndoB-GFP* | This paper | | Created by injecting pJFRC7-EndoB-GFP DNA into flies with the attP2 docking site |
| Genetic reagent (*D. melanogaster*) | *UAS-Cip4-GFP* | PMID: 19716703 | FLYB:FBtp0055642 | FlyBase symbol: P{UAS-Cip4.EGFP} |
| Genetic reagent (*D. melanogaster*) | *UAS-Arm-GFP* | Bloomington Drosophila Stock Center | BDSC:58725; FLYB:FBti0164941 | FlyBase symbol: M{UASp-arm. mGFP6}ZH-86Fb |
| Genetic reagent (*D. melanogaster*) | *UAS-shg-GFP* | Bloomington Drosophila Stock Center | BDSC:58445; FLYB:FBti0164810 | FlyBase symbol: P{UASp-shg.GFP}5B |
| Genetic reagent (*D. melanogaster*) | *UAS-baz-GFP* | Bloomington Drosophila Stock Center | BDSC:65845; FLYB:FBti0183077 | FlyBase symbol: P{UASp-baz.C.GFP}attP2 |
| Genetic reagent (*D. melanogaster*) | *UAS-Dlg-GFP* | Bloomington Drosophila Stock Center | BDSC:8610; FLYB:FBti0058796 | FlyBase symbol: P{UAS-Dl::GFP}DA53 |
| Genetic reagent (*D. melanogaster*) | *edGFP exon trap* | Kyoto Drosophila Genomics and Genetics Resources | Kyoto:115114; FLYB:FBti0144023 | FlyBase symbol: PBac{602 .P. SVS-1}edCPTI000616 |
| Genetic reagent (*D. melanogaster*) | *Nrg-GFP* | Bloomington Drosophila Stock Center | BDSC:6844; FLYB:FBti0027855 | FlyBase symbol: P{PTT-GA} NrgG00305 |
| Genetic reagent (*D. melanogaster*) | *NrxIV-GFP* | Bloomington Drosophila Stock Center | BDSC:50798; FLYB:FBti0099828 | FlyBase symbol: P{PTT-GA} Nrx-IVCA06597 |
| Genetic reagent (*D. melanogaster*) | *UAS-actin-GFP* | Bloomington Drosophila Stock Center | BDSC:9258; FLYB:FBti0072618 | FlyBase symbol: P{UASp-GFP.Act5C}2–1 |
| Genetic reagent (*D. melanogaster*) | *UAS-GMA-GFP* | Bloomington Drosophila Stock Center | BDSC:31776; FLYB:FBti0131132 | FlyBase symbol: P{UAS-GMA}3 |
| Genetic reagent (*D. melanogaster*) | *UAS-APC2-GFP* | Bloomington Drosophila Stock Center | BDSC:8815; FLYB:FBti0072923 | FlyBase symbol: P{UAS-Apc2.GFP}3 |
| Genetic reagent (*D. melanogaster*) | *UAS-LifeAct.mGFP* | Bloomington Drosophila Stock Center | BDSC:58717; FLYB:FBti0164965 | FlyBase symbol: M{UASp-LifeAct. mGFP6}ZH-2A |
| Genetic reagent (*D. melanogaster*) | *UAS-Arp3-GFP* | Bloomington Drosophila Stock Center | BDSC:39722; FLYB:FBti0148835 | FlyBase symbol: P{UASp-Arp3.GFP}3 |
| Genetic reagent (*D. melanogaster*) | *UAS-Arpc1-GFP* | Bloomington Drosophila Stock Center | BDSC:26692; FLYB:FBti0114937 | FlyBase symbol: P{UASp-Arpc1.GFP}1 |
| Genetic reagent (*D. melanogaster*) | *UAS-capu-GFP* | Bloomington Drosophila Stock Center | BDSC:24764; FLYB:FBti0100538 | FlyBase symbol: P{UASp-capu.GFP}20 |
| Genetic reagent (*D. melanogaster*) | *UAS-dpod1-GFP* | Bloomington Drosophila Stock Center | BDSC:8800; FLYB:FBti0072921 | FlyBase symbol: P{UAS-pod1.GFPmyc}3 |

*Continued*

| Reagent type | Designation | Source or reference | Identifiers | Additional information |
|---|---|---|---|---|
| Genetic reagent (*D. melanogaster*) | *UAS-Pak-GFP* | Bloomington Drosophila Stock Center | BDSC:52299; FLYB:FBti0155014 | FlyBase symbol: P{UASp-GFP-Pak1}300 |
| Genetic reagent (*D. melanogaster*) | *UAS-Rho1-GFP* | Bloomington Drosophila Stock Center | BDSC:9393; FLYB:FBti0074447 | FlyBase symbol: P{UASp-GFP.Rho1}6 |
| Genetic reagent (*D. melanogaster*) | *UAS-Rok-GFP* | Bloomington Drosophila Stock Center | BDSC:52290; FLYB:FBti0155000 | FlyBase symbol: P{UASp-Rok.RBD-GFP}30 |
| Genetic reagent (*D. melanogaster*) | *UAS-shot-GFP* | Bloomington Drosophila Stock Center | BDSC:29044; FLYB:FBti0127943 | FlyBase symbol: P{UAS-shot.L(A)-GFP}1 |
| Genetic reagent (*D. melanogaster*) | *UAS-GFP-blr* | Bloomington Drosophila Stock Center | BDSC:8659; FLYB:FBti0064611 | FlyBase symbol: P{UAS-GFP.BLR}3 |
| Genetic reagent (*D. melanogaster*) | *UAS-spir-GFP* | Bloomington Drosophila Stock Center | BDSC:8820; FLYB:FBti0072893 | FlyBase symbol: P{UAS-spir.L.GFP}3 |
| Genetic reagent (*D. melanogaster*) | *UAS-GFP-sstn* | Bloomington Drosophila Stock Center | BDSC:65863; FLYB:FBti0183072 | FlyBase symbol: P{UASp-GFP-sstn}attP40 |
| Genetic reagent (*D. melanogaster*) | *UAS-alpha-tubulin84B-GFP* | Bloomington Drosophila Stock Center | BDSC:7373; FLYB:FBti0038034 | FlyBase symbol: P{UASp-GFPS65C-αTub84B}3 |
| Genetic reagent (*D. melanogaster*) | *UAS-EB1-GFP* | Bloomington Drosophila Stock Center | BDSC:35512; FLYB:FBti0141213 | FlyBase symbol: P{UAS-EB1-GFP}3 |
| Genetic reagent (*D. melanogaster*) | *UAS-hook-GFP* | Bloomington Drosophila Stock Center | BDSC:65858; FLYB:FBti183087 | FlyBase symbol: P{UASp-hook.GFP}attP2 |
| Genetic reagent (*D. melanogaster*) | *UAS-jar-GFP* | Bloomington Drosophila Stock Center | BDSC:67606; FLYB:FBti0186539 | FlyBase symbol: P{UAS-jar.GFP}2 |
| Genetic reagent (*D. melanogaster*) | *UAS-Khc-GFP* | Bloomington Drosophila Stock Center | BDSC:9648; FLYB:FBti0076674 | FlyBase symbol: P{UAS-Khc.EGFP}2 |
| Genetic reagent (*D. melanogaster*) | *UAS-GFP-Myo10A* | Bloomington Drosophila Stock Center | BDSC:24781; FLYB:FBti0100569 | FlyBase symbol: P{UASp-Myo10A.GFP}20 |
| Genetic reagent (*D. melanogaster*) | *UAS-GFP-Myo31DF* | Bloomington Drosophila Stock Center | BDSC:1521; FLYB:FBti0003040 | FlyBase symbol: P{UAS-GFP.S65T} Myo31DFT2 |
| Genetic reagent (*D. melanogaster*) | *UAS-GFP-NinaC* | Bloomington Drosophila Stock Center | BDSC:43347; FLYB:FBti0151777 | FlyBase symbol: P{UAS-GFP-ninaC}2 |
| Genetic reagent (*D. melanogaster*) | *UAS-GFP-DCTN1-p150* | Bloomington Drosophila Stock Center | BDSC:29982; FLYB:FBti0128481 | FlyBase symbol: P{UAS-GFP.DCTN1-p150}2 |
| Genetic reagent (*D. melanogaster*) | *UAS-Unc104-GFP* | Bloomington Drosophila Stock Center | BDSC:24786; FLYB:FBti0100934 | FlyBase symbol: P{UAS-unc-104.GFP.RVB}1 |
| Genetic reagent (*D. melanogaster*) | *UAS-Supervillin-GFP* | Bloomington Drosophila Stock Center | BDSC:66165; FLYB:FBti0183648 | FlyBase symbol: P{UASp-GFP.Svil}attP2 |

*Continued*

| Reagent type | Designation | Source or reference | Identifiers | Additional information |
|---|---|---|---|---|
| Genetic reagent (*D. melanogaster*) | UAS-GFP-RhoGAP19D | Bloomington Drosophila Stock Center | BDSC:66167; FLYB:FBti0183650 | FlyBase symbol: P{UASp-GFP.RhoGAP19D}attP2 |
| Genetic reagent (*D. melanogaster*) | UAS-Par6-GFP | Bloomington Drosophila Stock Center | BDSC:65847; FLYB:FBti0183076 | FlyBase symbol: P{UAS-par-6.GFP}2 |
| Genetic reagent (*D. melanogaster*) | UAS-fy-GFP | Bloomington Drosophila Stock Center | BDSC:66513; FLYB:FBti0184640 | FlyBase symbol: P{UAS-fy.GFP}3 |
| Genetic reagent (*D. melanogaster*) | UAS-sds22.GFP | Bloomington Drosophila Stock Center | BDSC:65851; FLYB:FBti0183080 | FlyBase symbol: P{UASp-sds22.GFP}attP2 |
| Genetic reagent (*D. melanogaster*) | UAS-EGFP (cytosolic) | Bloomington Drosophila Stock Center | BDSC:5431; FLYB:FBti0013987 | FlyBase symbol: P{UAS-EGFP}5a.2 |
| Genetic reagent (*D. melanogaster*) | UAS-mito-GFP | Bloomington Drosophila Stock Center | BDSC:8443; FLYB:FBti0040804 | FlyBase symbol: P{UAS-mito-HA-GFP.AP}3 |
| Genetic reagent (*D. melanogaster*) | UAS-GFP-Golgi | Bloomington Drosophila Stock Center | BDSC:31422; FLYB:FBti0129989 | FlyBase symbol: P{UASp-GFP.Golgi}14 |
| Genetic reagent (*D. melanogaster*) | UAS-GFP-KDEL | Bloomington Drosophila Stock Center | BDSC:9898; FLYB:FBti0076567 | FlyBase symbol: P{UAS-GFP.KDEL}11.1 |
| Genetic reagent (*D. melanogaster*) | UAS-GFP.SKL | Bloomington Drosophila Stock Center | BDSC:28881; FLYB:FBti0127932 | FlyBase symbol: P{UAS-GFP.SKL}2 |
| Genetic reagent (*D. melanogaster*) | UAS-eGFP-Atg5 | Bloomington Drosophila Stock Center | BDSC:59848; FLYB:FBti0072907 | FlyBase symbol: P{UASp-eGFP-drAtg5}16 |
| Genetic reagent (*D. melanogaster*) | UAS-Atg8-GFP | Bloomington Drosophila Stock Center | BDSC:52005; FLYB:FBti0154554 | FlyBase symbol: P{UAS-Atg8a.GFP}2 |
| Genetic reagent (*D. melanogaster*) | UAS-GFP-LAMP | Bloomington Drosophila Stock Center | BDSC:42714; FLYB:FBti0150347 | FlyBase symbol: P{UAS-GFP-LAMP}2 |
| Genetic reagent (*D. melanogaster*) | UAS-Aplip-GFP | Bloomington Drosophila Stock Center | BDSC:24634; FLYB:FBti0100247 | FlyBase symbol: P{UASp-Aplip1.EGFP}3 |
| Genetic reagent (*D. melanogaster*) | UAS-hiw-GFP | Bloomington Drosophila Stock Center | BDSC:51640; FLYB:FBti0164779 | FlyBase symbol: P{UAS-GFP-hiw}B |
| Genetic reagent (*D. melanogaster*) | UAS-src-EGFP | Bloomington Drosophila Stock Center | BDSC:5429; FLYB:FBti0013989 | FlyBase symbol: P{UAS-srcEGFP}M7A |
| Genetic reagent (*D. melanogaster*) | UAS-bsk-GFP | Bloomington Drosophila Stock Center | BDSC:59267; FLYB:FBti0166949 | FlyBase symbol: P{UAS-bsk.GFP}2 |
| Genetic reagent (*D. melanogaster*) | UAS-Dronc-GFP | Bloomington Drosophila Stock Center | BDSC:56759; FLYB:FBti0161295 | FlyBase symbol: P{UAS-Dronc.EGFP}2 |
| Genetic reagent (*D. melanogaster*) | UAS-Myc-fry-GFP | Bloomington Drosophila Stock Center | BDSC:32106; FLYB:FBti0131207 | FlyBase symbol: P{UAS-Myc-fry-GFP}2 |
| Genetic reagent (*D. melanogaster*) | UAS-Fak-GFP | Gift. PMID: 15525665 | FLYB:FBtp0020317 | FlyBase symbol: P{UAS-Fak.EGFP} |

*Continued on next page*

*Continued*

| Reagent type | Designation | Source or reference | Identifiers | Additional information |
|---|---|---|---|---|
| Genetic reagent (*D. melanogaster*) | *UAS-muskelin-GFP* | Bloomington Drosophila Stock Center | BDSC:65860; FLYB:FBti0183089 | FlyBase symbol: P{UASp-muskelin. GFP}attP2 |
| Genetic reagent (*D. melanogaster*) | *UAS-roc2-GFP* | Bloomington Drosophila Stock Center | BDSC:65861; FLYB:FBti0183090 | FlyBase symbol: P{UASp-Roc2.GFP}attP2 |
| Genetic reagent (*D. melanogaster*) | *UAS-GFP-cora1-383* | This paper | | Created by injecting *UAS-GFP-cora1-383* DNA into flies with the VK00027 attP docking site |
| Strain, strain background (*Danio rerio*) | *TgBAC(tp63: GAL4FF)la213* | PMID: 25589751 | RRID:ZFIN_ZDB-ALT-150424-4 | NA |
| Strain, strain background (*Danio rerio*) | *Tg(isl1[ss]: LEXA-VP16,LEXAop: tdTomato)la215* | PMID: 25589751 | RRID:ZFIN_ZDB-ALT-150424-6 | NA |
| Strain, strain background (*Danio rerio*) | *Tg(isl1:GAL4-VP16, UAS:EGFP)zf154* | PMID: 15886097 | RRID:ZFIN_ZDB-ALT-090917-1 | NA |
| Strain, strain background (*Danio rerio*) | *Tg(isl1:GAL4-VP16 ,UAS:RFP)zf234* | PMID: 19962310 | RRID:ZFIN_ZDB-ALT-110520-2 | NA |
| Strain, strain background (*Danio rerio*) | *Tg(UAS:EGFP-PH-PLC)la216* | This paper | NA | Created by co-injection of pDEST-4xUASnr-EGFP-PH-PLC-pA and tol2 mRNA |
| Strain, strain background (*Danio rerio*) | *Tg(UAS:lifeact -GFP)mu271* | PMID: 23698350 | RRID:ZFIN_ZDB-ALT-130624-2 | NA |
| Strain, strain background (*Danio rerio*) | *Tg(UAS:GFP-CAAX)pd1025* | PMID: 23460678 | RRID:ZFIN_ZDB-ALT-130409-2 | NA |
| Strain, strain background (*Danio rerio*) | *Gt(jupa-citrine) ct520a* | PMID: 22056673 | RRID:ZFIN_ZDB-ALT-170123-1 | NA |
| Strain, strain background (*Danio rerio*) | *Gt(ctnna-citrine)ct3a* | PMID: 22056673 | RRID:ZFIN_ZDB-ALT-111010-23 | NA |
| Strain, strain background (*Danio rerio*) | *Gt(cdh1-tdtomato)xt18* | PMID: 30504889 | NA | NA |
| Strain, strain background (*Danio rerio*) | *AB (Wild-Type)* | Other | NA | Sagasti Lab stock |
| Antibody | Rabbit polyclonal anti-GFP antibody | Thermo Fisher Scientific | Thermo Fisher Scientific: A-11122; RRID:AB_221569 | 1:500; overnight at 4°C |
| Antibody | Mouse anti-GFP monoclonal antibody clone 3E6 | Thermo Fisher Scientific | Thermo Fisher Scientific: A-11120; RRID:AB_221568 | 1:500; overnight at 4°C |
| Antibody | Rabbit polyclonal anti-dsRed antibody | Clontech | Clontech Cat# 632496; RRID:AB_10013483 | 1:500; overnight at 4°C |
| Antibody | Mouse anti-coracle monoclonal antibody | Developmental Studies Hybridoma Bank | DSHB Cat# c566.9; RRID:AB_1161642 | 1:25; overnight at 4°C |

*Continued on next page*

*Continued*

| Reagent type | Designation | Source or reference | Identifiers | Additional information |
|---|---|---|---|---|
| Antibody | Goat anti-mouse IgG (H + L) cross-adsorbed secondary antibody, Alexa Fluor 568 | Thermo Fisher Scientific | Thermo Fisher Scientific: A-11004; RRID:AB_2534072 | 1:500; 2 h at 23°C |
| Antibody | Goat anti-rabbit IgG (H + L) cross-adsorbed secondary antibody, Alexa Fluor 555 | Thermo Fisher Scientific | Thermo Fisher Scientific: A-21428; RRID:AB_2535849 | 1:100; 4 h at 23°C |
| Antibody | Goat anti-mouse IgG (H + L) cross-adsorbed secondary antibody, Alexa Fluor 488 | Thermo Fisher Scientific | Thermo Fisher Scientific: A-31561; RRID:AB_2536175 | 1:100; 4 h at 23°C |
| Antibody | Goat anti-rabbit IgG (H + L) cross-adsorbed secondary antibody, Alexa Fluor 488 | Thermo Fisher Scientific | Thermo Fisher Scientific: A-11034; RRID:AB_2576217 | 1:100; 4 h at 23°C |
| Antibody | Donkey anti-rabbit IgG (H + L), ATTO 565-conjugated | This study | NA | 1:10; 4 h at 23°C |
| Antibody | Goat anti-horseradish peroxidase IgG, affinity purified Cy5 conjugate | Jackson Immunoresearch | Jackson Immunoresearch: 123-175-021 | |
| Recombinant DNA reagent | pJFRC7 | Addgene; PMID: 20697123 | Addgene: 26220 | |
| Recombinant DNA reagent | pJFRC-MUH | Addgene; PMID: 20697123 | Addgene: 26213 | |
| Recombinant DNA reagent | pUAST-EndoA-GFP | This paper | | Assembled using restriction enzyme digest |
| Recombinant DNA reagent | pUAST-EndoB-GFP | This paper | | Assembled using restriction enzyme digest |
| Recombinant DNA reagent | pJFRC-MUH-GFP-cora1-383 | This paper | | Assembled using restriction enzyme digest |
| Recombinant DNA reagent | pAA173 | PMID: 18190904 | NA | NA |
| Recombinant DNA reagent | p5E-krtt1c19e | PMID: 25589751 | NA | NA |
| Recombinant DNA reagent | p5E-4xUASnr | PMID: 21223961 | NA | NA |
| Recombinant DNA reagent | pME-EGFP | PMID: 17937395 | NA | NA |
| Recombinant DNA reagent | pME-EGFP-CAAX | PMID: 17937395 | NA | NA |
| Recombinant DNA reagent | pME-EGFP-PH-PLC | This paper | NA | Assembled using restriction enzyme digest |
| Recombinant DNA reagent | p3E-pA | PMID: 17937395 | NA | NA |

*Continued on next page*

*Continued*

| Reagent type | Designation | Source or reference | Identifiers | Additional information |
|---|---|---|---|---|
| Recombinant DNA reagent | *pDEST-4xUASnr-EGFP-PH-PLC-pA* | This paper | NA | Assembled using multisite gateway cloning |
| Recombinant DNA reagent | *pDEST-krtt1c19e-EGFP-CAAX-pA* | This paper | NA | Assembled using multisite gateway cloning |
| Recombinant DNA reagent | *pDEST-neurod 5 kb:mTangerine* | Other | NA | Gift from Alex Nechiporuk |
| Recombinant DNA reagent | pGFP-FRT-Kan-FRT | PMID: 22134125 | NA | NA |
| Recombinant DNA reagent | CH73-316A13 | BACPAC Resources Center | NA | NA |
| Recombinant DNA reagent | CH211-120J4 | BACPAC Resources Center | NA | NA |
| Recombinant DNA reagent | *dscl2-gfp* BAC | This paper | NA | Assembled using BAC recombineering |
| Recombinant DNA reagent | *dspa-gfp* BAC | This paper | NA | Assembled using BAC recombineering |
| Sequence-based reagent | Control morpholino | GeneTools | NA | NA |
| Sequence-based reagent | *neurog1* morpholino | GeneTools. PMID: 12413897, 12015292 | NA | NA |
| Chemical compound, drug | PBP10 | Sigma | Sigma Cat# 529625 | |
| Chemical compound, drug | AG1478 | Calbiochem | Calbiochem Cat#:658552 | NA |
| Software, algorithm | FIMTrack | PMID: 28493862 | | https://www.uni-muenster.de/PRIA/en/FIM/ |
| Software, algorithm | Ctrax | PMID: 19412169 | | http://ctrax.sourceforge.net/ |
| Software, algorithm | FIJI | PMID: 22743772 | Fiji, RRID:SCR_002285 | https://fiji.sc/ |
| Software, algorithm | R | R Project for Statistical Computing | R Project for Statistical Computing, RRID:SCR_001905 | https://www.R-project.org |

## Drosophila strains

Flies were maintained on standard cornmeal-molasses-agar media and reared at 25° C under 12 hal-ternating light-dark cycles. Alleles used in this study are detailed in the Key Resources Table, results from the reporter screen are detailed in *Supplementary file 1* and *Figure 1—figure supplement 1*. Experimental genotypes are listed in *Supplementary file 2*.

## Generation of transgenic fly lines
### UAS-EndoA-GFP and UAS-EndoB-GFP
We PCR-amplifed EndoA and EndoB from genomic DNA with the following primers:

> EndoA Forward: 5'-GAAGCGGCCGCTGGAAAAAATCGAAGATTAC-3'
> EndoA Reverse: 5'-GAAGGATCCGTTGCCATTGGGCAGGGGC-3'
> EndoB Forward: 5'-GGAGCGGCCGCTACGAAGTAGAGGAAATAAA-3'
> EndoB Reverse: 5'-CACGGATCCGAGGGTGACATCGTGCTCT-3'

PCR amplicons were ligated into pJFRC following NotI/BamHI digestion. The resulting plasmids (pJFRC7-EndoA and pJFRC7-EndoB) were used for phiC31 integrase-mediated transformation of flies carrying the *attP2* third chromosome attP docking site.

### UAS-GFP-cora[1-383]

We PCR-amplifed the amino terminal portion of the cora gene, which is sufficient to direct junctional localization of the gene product (Ward et al., 1998), using the following primers: Forward: 5'-GGAACTAGTATGCCGGCGGAAATTAAAC-3'; Reverse: 5'-GAACTCGAGCTACTTCTCCTTCTTG TTCTTGATGG-3'. The cora PCR amplicon was digested with SpeI and XhoI and ligated into a version of pJFRC-MUH (Pfeiffer et al., 2010) that was modified to contain the coding region of GFP as an in-frame amino-terminal fusion protein. The resulting plasmid (pJFRC-MUH-GFP-cora[1-383]) was used for phiC31 integrase-mediated transformation of flies carrying the VK00027 third chromosome attP docking site (Bestgene).

### PBP10 feeding

Larvae were transferred at 72 h AEL to 35 mm dishes containing unmodified cornmeal-molasses agar (mock) or cornmeal-molasses agar supplemented with 20 µM PBP10 and assayed for behavior responses and cora immunoreactivity at 120 h AEL.

### Zebrafish

Zebrafish (Danio rerio) were grown at 28.5°C on a 14 h/10 h light/dark cycle. The following previously described transgenic strains were used: TgBAC(tp63:GAL4FF)[la213], Tg(isl1[ss]: LEXA-VP16,LEX-Aop:tdTomato)[la215] (Rasmussen et al., 2015), Tg(isl1:GAL4-VP16,UAS:EGFP)[zf154] (Sagasti et al., 2005), Tg(isl1:GAL4-VP16,UAS:RFP)[zf234] (O'Brien et al., 2009a), Gt(ctnna-citrine)[ct3a] (Trinh et al., 2011), Gt(jupa-citrine)[ct520a] (Trinh et al., 2011), Tg(UAS:lifeact-GFP)[mu271] (Helker et al., 2013), Gt (cdh1-tdtomato)[xt18] (Cronan et al., 2018), and Tg(UAS:GFP-CAAX)[pd1025] (Ellis et al., 2013). All experimental procedures were approved by the Chancellor's Animal Research Care Committee at UCLA.

### BAC modification

To generate BAC reporters for dsc2l and dspa, the corresponding stop codons in BACs CH73-316A13 and CH211-120J4, respectively, were replaced by a GFP-KanR cassette as previously described (Suster et al., 2011).

### Zebrafish plasmid assembly

To generate pME-EGFP-PH-PLC, the PH domain of rat PLC1δ1 was PCR amplified from pAA173 (Kachur et al., 2008) and cloned into pME-EGFP (Kwan et al., 2007) using the restriction enzymes XhoI and BglII. The pDEST-4xUASnr-EGFP-PH-PLC-pA plasmid was created by Gateway cloning of p5E-4xUASnr (Akitake et al., 2011), pME-EGFP-PH-PLC, and p3E-pA (Kwan et al., 2007). pDEST-krtt1c19e-EGFP-CAAX-pA was assembled by Gateway cloning of p5E-krtt1c19e (Rasmussen et al., 2015), pME-EGFP-CAAX, and p3E-pA (Kwan et al., 2007).

### UAS:GFP-PH-PLC zebrafish transgenic line construction

To create a stable line, one cell stage embryos were injected with pDEST-4xUASnr-EGFP-PH-PLC-pA and tol2 mRNA, raised to adulthood and screened for transgene transmission to the F1 generation.

### Zebrafish transient transgenesis

To label lateral line axons, one to four-cell stage zebrafish embryos were injected with 25 pg of a neurod:mTangerine plasmid (gift from Alex Nechiporuk, Oregon Health and Science University, Portland, OR). 200 pg of BAC reporters for dsc2l and dspa were injected at the one to four-cell stage.

### Morpholino injection

To block somatosensory neuron development, one cell stage embryos were injected with 1 nl of injection mixture containing an antisense morpholino oligonucleotide targeting neurog1 (5'-ACGA TCTCCATTGTTGATAACCTGG-3') at a concentration of 0.7 mM (Andermann et al., 2002; Cornell and Eisen, 2002). Loss of response to touch was monitored to confirm efficacy of the treatment. As a control, embryos were injected with 1 nl of an antisense morpholino that targets an

intron of the human beta-globin gene (5'-CCTCTTACCTCAGTTACAATTTATA-3') at a concentration of 0.7 mM. Antisense morpholino oligonucleotides were synthesized by GeneTools (Philomath, OR).

## AG1478 treatment

The ErbB receptor antagonist AG1478 was used to perturb repositioning of the pLLn below the epidermis (*Raphael et al., 2010*). Embryos were bathed in embryonic medium containing either 4 µM AG1478/1% DMSO or 1% DMSO as a control.

## Microscopy

### Imaging

*Drosophila* larvae were mounted in 90% glycerol under No. one coverslips and imaged using a Leica SP5 microscope with a 40 × 1.2 NA oil immersion lens. For time-lapse analysis, larvae were imaged at the indicated time, recovered to yeasted agar plates with vented lids, aged at 25 °C, and imaged again. Zebrafish embryos were mounted as described (*O'Brien et al., 2009b*). Confocal imaging was performed on an LSM 510 or 800 confocal microscope (Carl Zeiss).

### Laser ablation

Larvae were mounted in 90% glycerol under No. one coverslips, dendrites were imaged using a Leica SP8 2-photon microscope with a 20 × 1.0 NA water immersion lens at 2x magnification under low (<20%) laser power. Cells were ablated or dendrites were severed by focusing high laser output (>80%) on the nucleus or a ~ 2 micron dendrite segment (64x magnification ROI scan), respectively. Larvae were recovered to yeasted agar plates with vented lids, aged at 25 °C, and processed for live imaging or immunostaining at the indicated time. Zebrafish axons were severed using a 2-photon laser as previously described (*O'Brien et al., 2009b*).

### Drosophila immunostaining

Third instar larvae were pinned on a sylgard plate, filleted along the ventral midline, and pinned open. After removing the intestines, fat bodies, imaginal discs, and ventral nerve cord, fillets were fixed in PBS with 4% PFA for 15 min at room temperature, washed four times for 5 min each in PBS with 0.3% Tx-100 (PBS-Tx), blocked for 1 h in PBS-Tx +5% normal donkey serum, and incubated in primary antibody overnight at 4° C. Samples were washed four times for 5 min each in PBS-Tx, incubated in secondary antibody for 4 h at room temperature, washed four times for 5 min each in PBS-Tx, and stored in PBS prior to imaging. Antibody dilutions were as follows: rabbit anti-GFP (Fisher #A-11122, 1:500), mouse anti-coracle (DSHB, C566.9 supernatant, 1:25), rabbit anti-dsRed (Clonetech #632496, 1:200), HRP-Cy5 (Jackson Immunoresearch, 1:100), goat anti-mouse Alexa488 (Thermofisher A-11001, 1:200), goat anti-rabbit Alexa 488 (Thermofisher A-11034, 1:200), goat anti-rabbit Alexa 555 (Thermofisher A-21428, 1:200).

## Drosophila expansion microscopy

Immunostaining was as above with the following antibodies: mouse anti-GFP, clone 3E6 (Invitrogen #A11120, 1:100), rabbit anti-dsRed (Clonetech #632496, 1:50), goat anti-mouse Alexa488 (Thermofisher A31561, 1:100), donkey anti-rabbit ATTO 565 (Vaughan lab, 1:10). Following immunostaining, samples were mounted on lysine-coated #1.5 cover glass in polydimethylsiloxane wells and incubated in monomer solution (2 M NaCl, 8.625% sodium acrylate, 2.5% acrylamide, 0.15% bisacrylamide in PBS) for 1 h at 4° C prior to gelation. A stock of 4-hydroxy-2,2,6,6-tetramenthylpiperidin-1-oxyl (4-hydroxy-TEMPO) at 1% (wt/wt) in water was added to the incubation solution and diluted to a concentration of 0.01%. Concentrated stocks of tetramethylethylenediamine (TEMED) and ammonium persulfate (APS) at 10% (wt/wt) in water were added sequentially to the incubation solution and diluted to concentrations of 0.2% (wt/wt). The tissues were then incubated at 37°C for 3–4 h. After gelation, the gels were cut and placed in a small 12-well chamber and 1 unit/ml (5 mg/ml) of chitinase in PBS (pH 6.0) was used to digest the cuticles for ~4 d at 37°C. Chitinase-treated samples were incubated with 1000 units/ml collagenase solution (prepared with buffer 1x HBSS lacking calcium, magnesium, and phenol red) with 0.01 M $CaCl_2$ and 0.01 M $MgCl_2$ overnight in a 37°C shaking incubation chamber. Samples were then rinsed with PBS twice for 5 min and digested in 8 units/ml proteinase K solution in digestion buffer (40 mM Tris pH 8.0, 1 mM EDTA, 0.5% Triton, 0.8 M

Guanidine HCl) for 1 h at 37°C. Subsequently, samples were removed from the digestion solution and were allowed to expand overnight in a large excess of deionized water. After expansion, the expanded gel was trimmed to fit onto the coverglass, excess water was removed, and the gel was mounted on a lysine-coated cover glass for imaging. Confocal microscopy was performed on a Leica SP5 inverted confocal scanning microscope using a 63 × 1.2 NA water lens.

## Drosophila SBF-SEM

Third instar larva were perforated with insect pins and cut open on ice in freshly made fixative (2.5% glutaraldehyde, 4% paraformaldehyde, 0.1 M sodium cacodylate). Samples were centrifuged at 15000 x rpm in a microcentrifuge for 1 h and then incubated at 4° C overnight to achieve thorough fixation. Next, samples were washed five times for 5 min each in 0.1 M sodium cacodylate and then post-fixed in osmium ferrocyanide for 1 h on ice. The tissues were then washed five times for 5 min each in $ddH_2O$ at room temperature and incubated in a 1% thiocarbohydrazide solution for 20 min at room temperature. The samples were washed five times for 5 min each in $ddH_2O$ at room temperature and then incubated in 2% osmium tetroxide for 30 min at room temperature. Following another five washes for 5 min each in $ddH_2O$ at room temperature, samples were stained en bloc in 1% uranyl acetate at 4° C overnight. The following day, tissues were washed five times for 5 min each in $ddH_2O$ at room temperature and stained en bloc in Walton's lead aspartate for 30 min at 60° C. The samples were then washed five times for 5 min each in $ddH_2O$ and dehydrated in an ice cold ethanol series (30%, 50%, 70%, and 95% EtOH), then transferred to room temperature for 5 min. This was followed by two changes of 100% EtOH and two changes of propylene oxide for 5 min each. The tissues were then infiltrated in a 1:1 mixture of propylene oxide: Durcupan resin, for 2 h at room temperature followed by overnight infiltration in fresh Durcupan. The following day, tissues were given a fresh change of Durcupan for 2 h at room temperature and then placed in flat embedding molds and polymerized in a 60° C oven for 2 days. The blocks were trimmed and imaged using a Zeiss Sigma scanning electron microscope with a Gatan 3-view system at 2.5–1.7 KV. Stacks (1000 sections) were collected with a 60 nm step size.

## Morphometric analysis

All image analysis was performed using Fiji (*Schindelin et al., 2012*). The Simple Neurite Tracer plugin (*Longair et al., 2011*) was used to trace neurites, ensheathment channels, and cell borders. Only basal cells for which the entire perimeter of the cell was visible were traced. R (https://www.r-project.org/) was used to generate plots and perform statistical tests.

## Behavior assays

*Harsh Touch*. Larvae were placed in a plastic petri dish with enough water, so larvae remained moist, but did not float in the dish. von Frey filaments made from fishing line and affixed to glass capillaries were applied to the dorsal side of the larvae between segments A3 and A6 until the filament buckled, exhibiting a pre-determined force (~78 mN). A positive response was scored if one complete nocifensive roll occurred within 10 s of the mechanical stimulus.

### Larval locomotion

Larvae were washed and placed on a 2% agar plate. To measure crawling velocity, 10 s videos of individual crawling larvae were recorded as uncompressed avi files using a Leica DFC310 FX camera on an AmScope FMA050 mount. Files were converted to flymovieformat with any2ufmf and analyzed in Ctrax (*Branson et al., 2009*). To measure crawling trajectory, larval locomotion was analyzed using the frustrated total internal reflection-based imaging method FIM together with the FIMTrack software package (*Risse et al., 2013*).

## Experimental design and statistical analysis

Datasets were tested for normality using Shapiro-Wilks goodness of fit tests. Details on statistical tests are provided in figure legends. Sample genotypes were blinded for both data acquisition and analysis.

## Acknowledgements

This work was supported by grants from the National Institutes of Health To JZP (NINDS R01 NS076614), AS (NIAMS R01 AR064582), JPR (NICHD K99 HD086271), and JCV (NIMH R01 MH115767), and to the UW Vision Research Core (NEI P30EY001730); a JSPS long-term fellowship and startup funds from UW to JZP; a WRF-Hall fellowship to KPL; a Jane Coffin Childs Memorial Fund fellowship to JPR. JPR is a WRF distinguished investigator. Fly Stocks obtained from the Bloomington *Drosophila* Stock Center (NIH P40OD018537) and antibodies obtained from the Developmental Studies Hybridoma bank, created by the NICHD of the NIH and maintained at The University of Iowa, were used in this study. We thank Julie Brill, Kazuo Emoto, David Parichy, and Peter Soba for helpful discussions, Le Trinh and Michel Bagnat for fish lines, Jeffrey Rosa for sharing images, Vasudha Chauhan for BAC cloning, and Stephen Basenfelder and Son Giang for excellent fish care.

## Additional information

### Funding

| Funder | Grant reference number | Author |
| --- | --- | --- |
| National Institute of Mental Health | NIMH R01 MH115767 | Joshua C Vaughan |
| National Institute of Neurological Disorders and Stroke | NINDS R01 NS076614 | Jay Z Parrish |
| National Institute of Arthritis and Musculoskeletal and Skin Diseases | NIAMS R01 AR064582 | Alvaro Sagasti |
| Eunice Kennedy Shriver National Institute of Child Health and Human Development | NICHD K99 HD086271 | Jeffrey P Rasmussen |
| Jane Coffin Childs Memorial Fund for Medical Research | | Jeffrey P Rasmussen |
| University of Washington | WRF-Hall fellowship | Kory P Luedke |
| Japan Society for the Promotion of Science | Long term fellowship | Jay Z Parrish |
| National Eye Institute | NEI P30EY001730 | Edward D Parker |

The funders had no role in study design, data collection and interpretation, or the decision to submit the work for publication.

### Author contributions

Nan Jiang, Conceptualization, Formal analysis, Investigation; Jeffrey P Rasmussen, Conceptualization, Formal analysis, Funding acquisition, Investigation, Writing—original draft, Writing—review and editing; Joshua A Clanton, Marci F Rosenberg, Investigation; Kory P Luedke, Investigation, Acquisition of Drosophila behavior data; Mark R Cronan, Resources, Contributed transgenic zebrafish lines; Edward D Parker, Investigation, Acquisition of Drosophila SEM data; Hyeon-Jin Kim, Investigation, Acquisition of Expansion Microscopy data; Joshua C Vaughan, Funding acquisition, Methodology, Supervision of expansion microscopy experiments; Alvaro Sagasti, Conceptualization, Formal analysis, Funding acquisition, Investigation, Methodology, Writing—original draft, Writing—review and editing, Supervision of zebrafish studies; Jay Z Parrish, Conceptualization, Data curation, Formal analysis, Funding acquisition, Investigation, Methodology, Writing—original draft, Project administration, Writing—review and editing, Supervision of Drosophila studies

### Author ORCIDs

Jeffrey P Rasmussen (iD) http://orcid.org/0000-0001-6997-3773
Joshua C Vaughan (iD) http://orcid.org/0000-0002-6550-8935
Alvaro Sagasti (iD) http://orcid.org/0000-0002-6823-0692
Jay Z Parrish (iD) http://orcid.org/0000-0002-0656-9148

### Ethics

Animal experimentation: All zebrafish procedures were approved by the Chancellor's Animal Research Care Committee at UCLA (protocol #2005-117-41C)

### Decision letter and Author response

Decision letter https://doi.org/10.7554/eLife.42455.029
Author response https://doi.org/10.7554/eLife.42455.030

## Additional files

### Supplementary files

• Supplementary file 1. Screen for markers associated with epidermal ensheathment channels. Related to *Figure 1*.
DOI: https://doi.org/10.7554/eLife.42455.025

• Supplementary file 2. Experimental *Drosophila* and zebrafish genotypes used in this study. Related to *Figures 1–7* and supplements.
DOI: https://doi.org/10.7554/eLife.42455.026

• Transparent reporting form
DOI: https://doi.org/10.7554/eLife.42455.027

### Data availability

All data generated or analysed during this study are included in the manuscript and supporting files.

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
