## [Decision Letter]

Thank you for submitting your article "A conserved morphogenetic mechanism for epidermal ensheathment of nociceptive sensory neurites" for consideration by *eLife*. Your article has been reviewed by two peer reviewers, and the evaluation has been overseen by a Reviewing Editor and Marianne Bronner as the Senior Editor. The following individual involved in review of your submission has agreed to reveal their identity: Yu Fengwei (Reviewer #2).

The reviewers have discussed the reviews with one another and the Reviewing Editor has drafted this decision to help you prepare a revised submission.

Summary:

In the manuscript by Jiang et al., the authors investigated a unique interaction between peripheral somatosensory neurites and epidermal tissues, namely neurite ensheathment. Although the structure and functions of the ensheathment have been described in several previous studies, this work provides a much more detailed picture of its molecular composition, formation, dynamics, and its roles in neurite branching morphogenesis and neuronal functions.

This work has several strengths. First, it consists of comparative studies in both *Drosophila* and zebrafish and therefore delineates evolutionarily conserved structures and mechanisms. Second, it provides the most comprehensive molecular characterization of the epidermal sheath to date and reveals important details of the signaling pathway in its formation and maintenance. Thirdly and most importantly, it addresses the direct effects of neurite ensheathment on neuronal morphogenesis and function, if the previously described roles in dendritic tiling and neuronal territory sharing can be considered indirect or secondary. Overall, this study reports interesting novel findings and reveals perhaps previously underappreciated importance of a specific interaction between sensory neurons and epidermal cells.

Major concerns:

In spite of our interest in your work, the manuscript in its current form has several major issues that reduce the significance of the manuscript. I would be much more enthusiastic about supporting its publication in *eLife* if these issues can be addressed.

1) Throughout the manuscript, PLC^δ^-PH-GFP was used as a marker to indicate epidermal sheaths. However, whether PLC^δ^-PH-GFP enrichment always correlates with neurite ensheathment has never been addressed. As an early event in epidermal sheath formation, PLC^δ^-PH-GFP could be recruited to the sites of epidermal plasma membrane opposing dendrites well before sheath formation. In fact by the authors' own admission, this seems to be the case for zebrafish sensory axons (Figure 5E). As the correlation between PLC^δ^-PH-GFP and epidermal sheaths is the basis of many conclusions, it needs to be validated. Since the authors have established expansion microscopy, one possible way is to conduct a more comprehensive survey of PLC^δ^-PH-GFP enrichment and epidermal sheaths (perhaps combined with an ECM staining). Another option is to compare extracellular HRP staining with PLC^δ^-PH-GFP pattern.

2) Jan and colleagues have previously reported another way dendrites can be embedded in epidermal cells without involving epidermal sheaths (Han et al. 2012). In that case, the dendrite are not connected to the basal epidermal surface through mesaxon-like structures. Although this type of ensheathment may represent a different class of structure with unique molecular composition and function and may not need to be the focus of this work, it should not be completely ignored. At the minimum, the authors should address how prevalent it is among ensheathed dendrites and how it may or may not be related to PLC^δ^-PH-GFP and other markers of epidermal sheaths. This can be addressed experimentally or by discussion.

3) The authors reported an increase of ensheathment during development and noted that "sheath formation is first initiated on long-lived dendrite shafts in proximal portions of the dendrite arbor rather than the more dynamic distal portions of the dendrite arbor". This is consistent with the idea that ensheathment is generated by the epidermal basal surface wrapping around stable dendrites. However, whether the dynamic part of dendrites gets wrapped was not directly addressed. The authors observed new branch formation on ensheathed dendrites (Figure 6P), but it wasn't made clear whether the new branches were ensheathed as well. It was also reported that ensheathed terminal dendrites can grow while remaining ensheathed (Han et al. 2012). How do epidermal sheaths change as a consequence? These questions should be addressed by time-lapse imaging.

4) The authors suggest a specific sequence of sheath assembly whereby PIP2 microdomains recruit other makers. If this is true, later markers (e.g. dArf6, F-actin, Cora) should always colocalize with PLC^δ^-PH. This seems to be what the manuscript implies but it was never clearly stated. In order to support this claim, direct evidence of recruitment of late markers to existing PLC^δ^-PH positive sheaths is needed.

---

## [Author Response]

Major concerns:In spite of our interest in your work, the manuscript in its current form has several major issues that reduce the significance of the manuscript. I would be much more enthusiastic about supporting its publication in eLife if these issues can be addressed.1) Throughout the manuscript, PLC^δ^-PH-GFP was used as a marker to indicate epidermal sheaths. However, whether PLC^δ^-PH-GFP enrichment always correlates with neurite ensheathment has never been addressed. As an early event in epidermal sheath formation, PLC^δ^-PH-GFP could be recruited to the sites of epidermal plasma membrane opposing dendrites well before sheath formation. In fact by the authors' own admission, this seems to be the case for zebrafish sensory axons (Figure 5E). As the correlation between PLC^δ^-PH-GFP and epidermal sheaths is the basis of many conclusions, it needs to be validated. Since the authors have established expansion microscopy, one possible way is to conduct a more comprehensive survey of PLC^δ^-PH-GFP enrichment and epidermal sheaths (perhaps combined with an ECM staining). Another option is to compare extracellular HRP staining with PLC^δ^-PH-GFP pattern.

We thank the reviewers for pointing out this important point that requires clarification. From our high-resolution analysis of epidermal PLC-PH-GFP distribution using ExM, we found that PLC-PH-GFP labeling at sites of neurite-epidermis contact corresponded to epidermal PLC-PH-GFP-positive membrane invaginations that ensheath neurites. Figures 1J-1K are representative of what we’ve seen with ExM analysis of 6 neurons; we more clearly state this in the figure legend of the revised manuscript.

To more systematically analyze whether epidermal PLC-PH-GFP labeling correlates with neurite ensheathment in *Drosophila*, we examined the relationship between neuronal HRP (a surface antigen) and epidermal PLC-PH-GFP signal intensity under non-permeabilizing staining conditions. We reasoned that if PLC-PH-GFP correlates with neurite ensheathment, HRP signal intensity (a proxy for exposed neurites) should be inversely related to intensity of PLC-PH-GFP labeling. This is precisely what we found, both at stages when PLC-PH-GFP initially accumulates adjacent to c4da dendrites (84 h AEL) and at a later stage (120 h) when large portions c4da dendrite arbors are ensheathed. We included these new results as Figure 1—figure supplement 2.

In addition to these new results, we modified the model (Figure 5I) to more accurately reflect the relative position of sheaths and the basal surface of epidermal cells.

*2) Jan and colleagues have previously reported another way dendrites can be embedded in epidermal cells without involving epidermal sheaths (Han et al. 2012). In that case, the dendrite are not connected to the basal epidermal surface through mesaxon-like structures. Although this type of ensheathment may represent a different class of structure with unique molecular composition and function and may not need to be the focus of this work, it should not be completely ignored. At the minimum, the authors should address how prevalent it is among ensheathed dendrites and how it may or may not be related to* PLC^δ^-PH-GFP *and other markers of epidermal sheaths. This can be addressed experimentally or by discussion.*

The prevalence of neurite embedding independent of ensheathment was not a major focus of this manuscript, so we cannot provide any direct experimental evidence that speaks to origin of such structures. In our SBF-SEM data we saw no evidence of these sheath-independent embedded dendrites, so we suspect that they are quite rare, but we cannot rule out the possibility that these structures are less efficiently labeled in our staining protocol. One possible source for these structures is remodeling/loss of PLC-PH-GFP sheaths following neurite embedding. Our time-lapse analysis demonstrates that retraction/loss of PLC-PH-GFP sheaths is a rare event, which could explain why we did not frequently detect these structures. We added a brief treatment of these structures to the Discussion in our revised manuscript.

3) The authors reported an increase of ensheathment during development and noted that "sheath formation is first initiated on long-lived dendrite shafts in proximal portions of the dendrite arbor rather than the more dynamic distal portions of the dendrite arbor". This is consistent with the idea that ensheathment is generated by the epidermal basal surface wrapping around stable dendrites. However, whether the dynamic part of dendrites gets wrapped was not directly addressed.

The reviewers identify an important point that we did not clearly address in the first submission. In the revised manuscript we added results that more directly speak to the relationship between ensheathment and dendrite dynamics. First, we measured the extent of terminal dendrite ensheathment and found that dendrite terminals are less extensively ensheathed than the dendrite arbor as a whole. Using time-lapse imaging, we monitored ensheathment status of newly formed terminal dendrites (or newly elongated portions of terminal dendrites) and found that new dendrite terminals are rarely ensheathed. We added these new results as Figure 6P and Figure 6—figure supplement 3.

Next, we monitored the frequency of new branch initiation from ensheathed neurites. To this end, we conducted time-lapse analysis of dendrite growth in larvae expressing epidermal sheath markers (Figure 6P, n=20 neurons total) and found that <9% of new branching events occur on ensheathed neurites (Figure 6Q). Most of these branching events occur at the “ends” of sheaths, so we suspect that mature sheaths are not permissive to dendrite branching. We added a more thorough discussion of the branching dynamics and a figure supplement displaying the relationship between ensheathment and new branching events, including one of the rare examples of new branch formation from an ensheathed dendrite (Figure 6—figure supplement 3).

The authors observed new branch formation on ensheathed dendrites (Figure 6P), but it wasn't made clear whether the new branches were ensheathed as well.

As described above, new branch formation on ensheathed dendrites is very rare. To ensure that we were not undersampling these events in our original dataset, we conducted additional time-lapse imaging to monitor the relationship between ensheathment and dendrite branching. Consistent with our prior results, we found that <9% of new branching events occurred on ensheathed portions of the dendrite arbor (updated results shown in Figure 6Q). Given the rarity of these events, we have not attempted to systematically address the ensheathment status of this specific subset of branches. However, our time-lapse analysis demonstrates that ensheathment of new branches is extremely rare (only ~5% of new terminals) and never included the entirety of the new terminal. These new results are depicted in Figure 6P and Figure 6—figure supplement 3.

It was also reported that ensheathed terminal dendrites can grow while remaining ensheathed (Han et al. 2012). How do epidermal sheaths change as a consequence? These questions should be addressed by time-lapse imaging.

Before comparing our results to those results reported in Han et al. 2012, it’s important to make the distinction between their “enclosed” dendrites, which were detached from the basement membrane, and our “ensheathed” dendrites, which are marked by epidermal PLC-PH-GFP-positive membrane domains that wrap the dendrites. We suspect that “enclosed” dendrites include the ensheathed dendrites that have monitored but may additionally include other populations of dendrites including dendrites that insert at epidermal cell-cell junctions, dendrites that transiently insert into epidermal cells without inducing sheath formation and/or pre-ensheathed dendrites that have not yet induced PLC-PH-GFP recruitment.

During the time window when Han et al. reported on dynamic growth of “enclosed” terminals (beginning at 72 h AEL), ensheathment as assessed by PLC-PH-GFP (<10% of the arbor) or coracle (<3% of the arbor) labeling is very rare and confined almost exclusively to dendrite shafts in the proximal portion of the dendrite arbor. We see no terminal dendrites that are wholly ensheathed by these markers at this early time point.

4) The authors suggest a specific sequence of sheath assembly whereby PIP2 microdomains recruit other makers. If this is true, later markers (e.g. dArf6, F-actin, Cora) should always colocalize with PLC^δ^-PH. This seems to be what the manuscript implies but it was never clearly stated. In order to support this claim, direct evidence of recruitment of late markers to existing PLC^δ^-PH positive sheaths is needed.

We thank the reviewers for raising this important point; the new results we added to the revision more clearly demonstrate the sequential recruitment of ensheathment factors. For these studies, we conducted time-lapse imaging of sheath maturation in larvae expressing PLC-PH-Cerulean together with other GFP-tagged sheath markers in the epidermis including dArf6-GFP, GMA-GFP, and GFP-cora (a new marker we generated for the revision to facilitate live-imaging of cora localization of sheaths). We found that epidermal sheaths wrapping c4da dendrites are first labeled by PLC-PH-Cerulean and then subsequently recruit other markers. Further, we found that PLC-PH-positive structures very rarely converted to PLC-PH-negative structures (consistent with our time-lapse imaging of sheath dynamics using PLC-PH-GFP), that double-positive sheaths remained double-positive, and that late sheath markers were never recruited in the absence of PLC-PH recruitment. We incorporated these new results into the revision Figure 5E-5F and Figure 5—figure supplement 1, and additionally expanded the corresponding portion of the text.